# Molecular basis of neurodegeneration in a mouse model of *Polr3*-related disease

Robyn D Moir[1]*[†], Emilio Merheb[1†], Violeta Chitu[2], E Richard Stanley[2], Ian M Willis[1,3]*

[1]Department of Biochemistry, Albert Einstein College of Medicine, Bronx, United States; [2]Department of Developmental and Molecular Biology, Albert Einstein College of Medicine, Bronx, United States; [3]Department of Systems and Computational Biology, Albert Einstein College of Medicine, Bronx, United States

## eLife Assessment

This study provides **important** insights into the mechanistic basis of neurological manifestations of RNA polymerase III-related disease by creating a mutant mouse to dissect transcriptional changes. The data provide **compelling** evidence for disease progression initiated by a global reduction in tRNA levels leading to integrated stress and innate immune responses and neuronal loss. The work will be of interest to those engaged in the study of chromosome biology, developmental biology and neurodegeneration.

**\*For correspondence:**
robyn.moir@einsteinmed.edu (RDM);
ian.willis@einsteinmed.edu (IMW)

[†]These authors contributed equally to this work

**Competing interest:** The authors declare that no competing interests exist.

**Abstract** Pathogenic variants in subunits of RNA polymerase (Pol) III cause a spectrum of *Polr3*-related neurodegenerative diseases including 4H leukodystrophy. Disease onset occurs from infancy to early adulthood and is associated with a variable range and severity of neurological and non-neurological features. The molecular basis of *Polr3*-related disease pathogenesis is unknown. We developed a postnatal whole-body mouse model expressing pathogenic *Polr3a* mutations to examine the molecular mechanisms by which reduced Pol III transcription results primarily in central nervous system phenotypes. *Polr3a* mutant mice exhibit behavioral deficits, cerebral pathology and exocrine pancreatic atrophy. Transcriptome and immunohistochemistry analyses of cerebra during disease progression show a reduction in most Pol III transcripts, induction of innate immune and integrated stress responses and cell-type-specific gene expression changes reflecting neuron and oligodendrocyte loss and microglial activation. Earlier in the disease when integrated stress and innate immune responses are minimally induced, mature tRNA sequencing revealed a global reduction in tRNA levels and an altered tRNA profile but no changes in other Pol III transcripts. Thus, changes in the size and/or composition of the tRNA pool have a causal role in disease initiation. Our findings reveal different tissue- and brain region-specific sensitivities to a defect in Pol III transcription.

## Introduction

Biallelic pathogenic mutations in multiple subunits of RNA polymerase (Pol) III are causally associated with a spectrum of neurodegenerative diseases (*Lata et al., 2021*; *Watt et al., 2023*). These *Polr3*-related disorders include a prevalent form of leukodystrophy with hypomyelination, hypodontia, and hypogonadotropic hypogonadism (4H leukodystrophy) as distinguishing features along with cerebellar atrophy, myopia and short stature (*Wolf et al., 2014*). Patients with *Polr3*-related leukodystrophy typically present in early childhood with motor deficits and developmental delay. The disease is inherited in an autosomal recessive manner, is progressive and exhibits wide ranging phenotypic

severity from premature death during infancy to mild forms that have been diagnosed in adults (*Watt et al., 2023*). The neurologic and non-neurologic manifestations of *Polr3*-related disorders are variably penetrant. Notably, initial findings of hypomyelination, suggesting oligodendrocyte involvement, are not universal as some patients have normal or near normal myelination but distinct MRI and neuropathological changes that suggest a largely neuronal phenotype (*Perrier et al., 2022*). Other atypical forms of *Polr3*-related disease have also been reported (*Lata et al., 2021*; *Watt et al., 2023*). The bases for these different disease presentations are unclear. It is also unclear why mutations in Pol III, a ubiquitously-expressed, essential enzyme responsible for the synthesis of critical non-coding RNAs central to protein synthesis, pre-mRNA splicing, secretion and other processes, result in phenotypes primarily of the central nervous system.

Phenotypic variability in *Polr3*-related disease is likely to reflect differences in the abundance of Pol III, its residual activity and how these changes affect the Pol III transcriptome at sensitive periods during embryonic and postnatal development. These properties are difficult to assess given the structural complexity of the 17 subunit Pol III enzyme and the fact that most patients carry compound heterozygous mutations that are likely to differentially affect enzyme function (*Wolf et al., 2014*). Genetic modifiers of *Polr3*-related phenotypes are also likely given the observed phenotypic variation between unrelated patients with the same *Polr3* mutation (*Bernard et al., 2011*; *Perrier et al., 2020*). Few studies have examined the effect of pathogenic *Polr3* mutations on the levels of Pol III-derived transcripts in mammalian cells. Collectively, experiments with patient-derived fibroblasts and CRISPR-Cas9 engineered cell lines show that precursor tRNA levels are generally lower but only one of four mature tRNAs tested in two different studies were reduced, suggesting that high tRNA stability can compensate for defects in synthesis (*Choquet et al., 2019*; *Dorboz et al., 2018*). 7SL RNA levels were uniformly lower in agreement with a separate analysis of Pol III-derived transcripts in patient blood (*Azmanov et al., 2016*). Other Pol III-transcribed RNAs (5 S RNA) showed discordant results or, based on one report, *U6*, *Y*, *Rpph1*, *Rmrp* and *7SK* RNAs were unchanged (*Choquet et al., 2019*). The extent to which the preceding findings reflect altered RNA levels in the brain is unknown.

In previous work we showed that brain-specific expression of pathogenic *Polr3a* mutations (W671R/G672E) in the mouse Olig2 lineage resulted in impaired growth and developmental delay, deficits in cognitive, sensory, and fine sensorimotor function, and hypomyelination in multiple regions of the cerebrum and spinal cord (*Merheb et al., 2021*). We report here a new model of *Polr3*-related disease in which expression of a Pol III enzyme bearing the *Polr3a* mutation is broadly induced postnatally in adolescent mice. Overt pathology was noted only in the exocrine pancreas and cerebrum despite similarly efficient recombination in multiple tissues. Analysis of the Pol III transcriptome reveals a decrease in pre-tRNA and mature tRNA populations and few if any changes among other Pol III transcripts across multiple tissues. Analysis of the Pol II transcriptome reveals activation of the integrated stress response in cerebra but not in other surveyed tissues. The molecular and cellular changes that underlie cerebral neurodegeneration in *Polr3a* mutant mice, which include induction of an innate immune response and activation of microglia, indicate different brain region-specific sensitivities to defective Pol III activity.

## Results

### A postnatal whole-body *Polr3a* mutant mouse

Patients with *Polr3*-related leukodystrophy express mutant forms of Pol III in all their cells. We sought to generate mice with a similarly broad cellular distribution of a Pol III enzyme bearing the *Polr3a* W671R/G672E mutation. To circumvent the embryonic lethality of the mutation, we employed a ubiquitously expressed tamoxifen-inducible CAGGCre-ER transgene to knockin the *Polr3a* mutant allele postnatally in diverse tissues (*Hayashi and McMahon, 2002*). Five tamoxifen injections were administered every other day starting at postnatal day 28 (P28, *Figure 1A and B*). Recombination frequencies determined by flow cytometry or fluorescence microscopy using a dual tdTomato-EGFP reporter were ~60% in liver, ~70% in cerebrum, ~80% in cerebellum and kidney and ~90% in heart (*Figure 1—figure supplement 1*). Tamoxifen-induced knockin mice (denoted as Polr3a-tamKI mice) have significantly lower body weights than tamoxifen-injected WT controls. Body weight differences appear in both sexes at P34 and become progressively larger as Polr3a-tamKI mice lose weight while the control mice gain weight over time (*Figure 1C*). These differences are due in part to the smaller

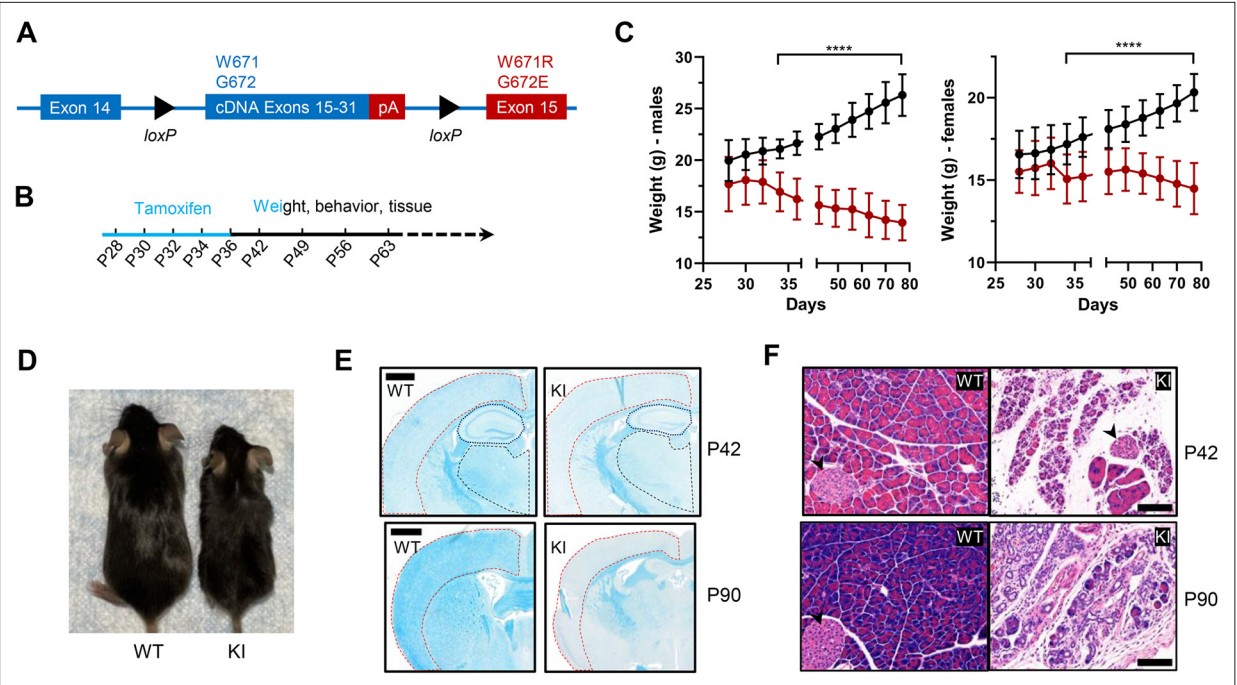

**Figure 1.** Body weight, length, and histological characteristics of Polr3a-tamKI mice. (**A**) Schematic of the modified Polr3a locus showing floxed WT sequences encoding exons 15–31 with adjacent sequences for termination and polyadenylation (pA) and Polr3a exon15 containing the leukodystrophy mutations. (**B**) Tamoxifen injections and experimental timeline. (**C**) Lower body weights of Polr3a-tamKI (red) versus WT (black) mice (males n=18/group, nested t test p<0.0001****, females n=20/group, nested t test p<0.0001****). Data show mean ± SD. (**D**) Body length differences at P56. Images are representative. (**E**) LFB-staining of WT and KI brain sections at P42 and P90. The cerebral cortex, hippocampus, and thalamus are marked by red, blue, and black lines, respectively. Scale bar, 1000 µm. (**F**) H&E-stained WT and KI pancreata at P42 and P90 shows a dramatic loss of acinar cells. Pancreatic islets are marked with a black arrowhead. Scale bar,100 µm.

The online version of this article includes the following figure supplement(s) for figure 1:

**Figure supplement 1.** Analysis of recombination frequency in Polr3a-tamKI mice.

**Figure supplement 2.** Cerebellar histology and glucose homeostasis of Polr3a-tamKI mice.

body size of Polr3a-tamKI mice post-injection (*Figure 1D*), suggesting a growth defect. Notably, a growth defect was reported previously for Polr3a-Olig2KI mice (formerly Polr3a-cKI mice, *Merheb et al., 2021*), consistent with the reduced stature of many patients with Pol III-related leukodystrophy (*Watt et al., 2023*). However, in contrast to the current model, Polr3a-Olig2KI mice gain weight, albeit more slowly than controls, from adolescence to adulthood despite their smaller size (*Merheb et al., 2021*). Thus, the loss of body weight in Polr3a-tamKI mice demonstrates a more severe failure to thrive phenotype, in line with the widespread expression of the *Polr3a* mutation.

## Impaired myelination and exocrine pancreatic atrophy

We conducted a broad histopathological analysis of the effect of the *Polr3a* mutation in adolescent (P42) and adult (P90) mice. Gross morphology was assessed by Hematoxylin and Eosin (H&E) staining and in the CNS, myelin and neurons were examined using Luxol Fast Blue (LFB) and Nissl, respectively. LFB staining of coronal sections of the cerebrum revealed impaired myelin deposition in Polr3a-tamKI cortex, hippocampus and thalamus with qualitatively greater differences in the cortex at P90 (*Figure 1E*). In contrast, LFB- and Nissl-stained sagittal sections of the cerebellum showed no differences in myelin or granular layer neurons, respectively, and no difference in Purkinje cell density (*Figure 1—figure supplement 2A, B*). The absence of a myelination defect in the cerebellum was previously noted in Polr3a-Olig2KI mice (*Merheb et al., 2021*). Non-CNS tissues (>20 examined) were histologically normal except for the pancreas. Polr3a-tamKI mice exhibit pronounced and severe exocrine pancreatic atrophy at P42 and P90, respectively (*Figure 1F*). This condition is characterized by insufficient production of digestive enzymes (*Kleeff et al., 2017*) and correlates with the lower body weight of Polr3a-tamKI mice, the marked reduction in subcutaneous and visceral adipose

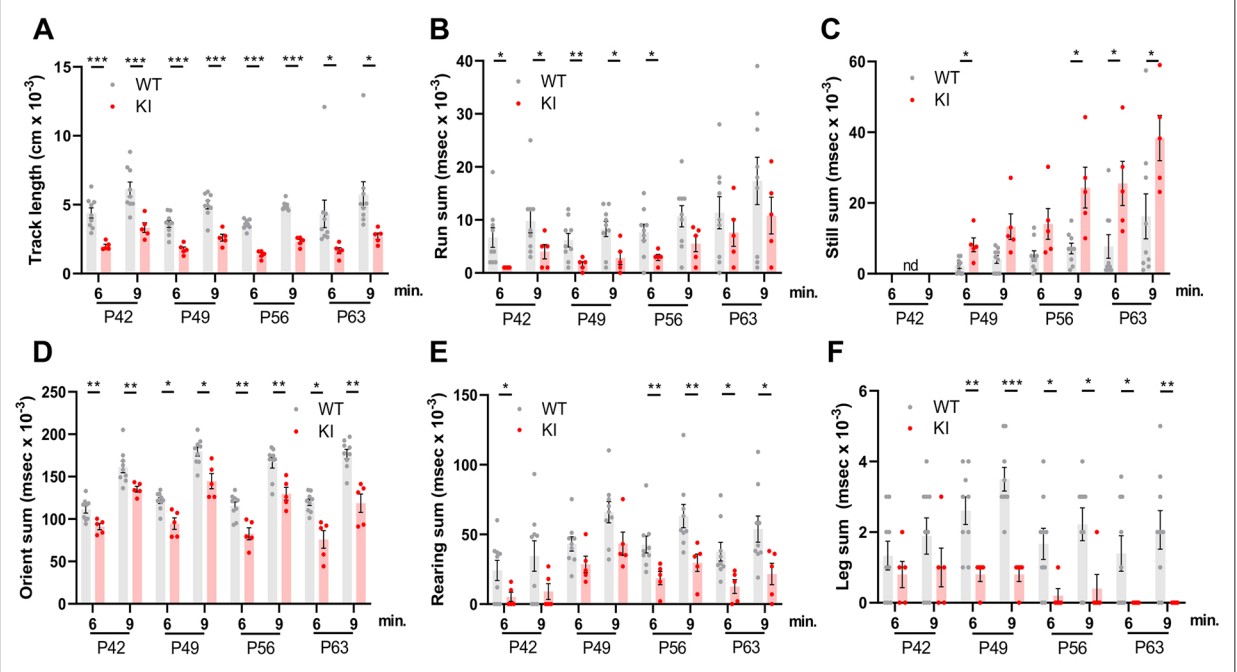

**Figure 2.** Behavioral studies of WT and Polr3a-tamKI mice. (**A–C**) Locomotor activity is reported as the total distance traveled (Track length) and the total time during which the mice are running (Run sum) or standing still (Still sum). (**D**) Risk assessment (Orient sum) is the sum of three orientation behaviors (see *Figure 2—figure supplement 1*). (**E**) Exploratory behavior (Rear sum) is the sum of three rearing behaviors (see *Figure 2—figure supplement 1*). (**F**) Leg grooming. Data were collected at weekly intervals beginning at P42 (WT n=9 and KI n=5). Mice were tested for nine minutes with recording in three intervals of 3 min each. Values show the mean ± SEM at 6- and 9-min timepoints. Multiple t tests and one-way ANOVA, * p<0.05, ** p<0.01, *** p<0.001.

The online version of this article includes the following figure supplement(s) for figure 2:

**Figure supplement 1.** Behavioral spectrometer analysis of WT and Polr3a-tamKI mice.

tissue at P90, and the poor health of the mice at this age. In contrast to the paucity of body fat at P90, echoMRI measurements at P42 showed that body fat as a percentage of total body weight was comparable to WT (*Figure 1—figure supplement 2C*). To assess whether the exocrine pancreatic defect might extend to the endocrine pancreas, glucose homeostasis was examined in adolescent mice. Blood glucose concentrations increased upon feeding, as expected, and were not significantly different between Polr3a-tamKI and WT mice in either the fasted or the refed state (*Figure 1—figure supplement 2D*). Similarly, a glucose tolerance test showed comparable responses (*Figure 1—figure supplement 2E*). These data suggest that endocrine pancreatic functions controlling glucose homeostasis are not significantly impaired at P42 by the *Polr3a* mutation.

## Polr3a-tamKI mice exhibit multiple behavioral changes

A behavioral spectrometer with video tracking and pattern-recognition software was used to quantify ~20 home cage-like behaviors in a longitudinal study of behavior. Testing began at P42 and was continued at weekly intervals (*Figure 1B*). No differences were observed in anxiety-like behavior, which was assessed using metrics that measure the natural tendency of mice to avoid open spaces (*Figure 2—figure supplement 1A–C*). However, overall locomotor activity was decreased in Polr3a-tamKI mice as indicated by the reduced track length at P42, P49, P56, and P63 (*Figure 2A*). Polr3a-tamKI mice also spent less time running and more time standing still at all time points compared to WT mice, consistent with a locomotor deficit (*Figure 2B and C*). Risk assessment was measured by orientation behavior according to three discrete metrics that can be summed to provide an overall evaluation of this behavior (*Brodkin et al., 2014*). Individual orientation behaviors were significantly lower in Polr3a-tamKI mice at all time points compared to WT mice, except for Orient-shuffle at P42 where there was a trend towards lower performance in the knockin mice (*Figure 2—figure supplement 1D–F*). The sum of these behaviors shows that Polr3a-tamKI mice spend less time assessing risk at all time points

(*Figure 2D*). Exploratory behavior was investigated via three rearing motions performed when the mice are standing on their hind legs. Polr3a-tamKI mice spent less time performing these motions at most timepoints and a tendency towards diminished performance at other times (*Figure 2—figure supplement 1G–I*). When all three rearing metrics were combined, Polr3a-tamKI mice showed lower levels of exploration compared to WT mice at all timepoints although the data did not reach significance at P49 (*Figure 2E*). Grooming behavior was evaluated by measuring the time each animal spent attending to a specific region of their body (*Figure 2F* and *Figure 2—figure supplement 1J, K*). The Polr3a-tamKI mice spent less time grooming their legs and back compared to WT mice starting at P49 and progressing to P63. Head grooming was also reduced at P63. In summary, the findings reveal behavioral phenotypes in Polr3a-tamKI mice at the earliest time point in the study, two weeks after introduction of the mutation, consistent with neurobehavioral deficits (*Brodkin et al., 2014*). As none of the affected behaviors were apparent in Polr3a-Olig2KI mice (*Merheb et al., 2021*), the data further demonstrate that the mutation has a more severe impact on animal behavior when its expression is induced postnatally in many cell types than when it is induced during embryonic development only in the Olig2 lineage.

## A reduction in Pol III transcripts in the cerebra of adult Polr3a-tamKI mice is accompanied by induction of innate immune and integrated stress responses

To investigate the molecular changes to Pol III transcript levels caused by the *Polr3a* mutation and any secondary effects on the Pol II transcriptome, we initially focused on the cerebra of adult mice at P75. Precursor tRNA levels reflect Pol III transcription activity since these nascent RNAs are rapidly processed into mature sized tRNA by the removal of 5' leader, 3' trailer and, in some instances, intronic sequences (*Berg and Brandl, 2021*). The levels of four precursor tRNAs, determined either by northern blotting for intron sequences or by RT-qPCR with precursor-specific primers, were reduced ~threefold in Polr3a-tamKI cerebra (*Figure 3A–C*). Mature tRNAs and other Pol III transcripts have relatively long half-lives and thus their levels reflect steady state abundance rather than synthesis (*Nwagwu and Nana, 1980*). Nonetheless, Northern blotting of four mature tRNAs showed that all were lower in Polr3a-tamKI cerebra with two of them reduced to ~60% of the WT (*Figure 3A and B*).

Despite the defect in precursor tRNA synthesis and the reduction in mature tRNAs, the Pol III transcriptome was not universally affected (*Figure 3A–C*). Transcripts from genes with a tRNA-like Type II promoter (e.g. *7SL*, *BC1* and *Mvg1*, aka *Vault* RNA) showed a decrease in abundance to between 45% and 60% of the WT level (*Figure 3C*). Transcripts from genes with a Type III promoter (e.g. *U6*, *Rmrp*, *Rpph1*, and *7SK* RNAs), were variably decreased: *Rpph1* and *7SK* RNAs were decreased to 70% and 56%, respectively, while *U6*, *U6atac* and *Rmrp* RNAs were unaffected. The level of 5S rRNA, the only RNA produced using a Type I promoter, was elevated (*Figure 3C*).

Mouse models of neurodegeneration with underlying defects in the translation machinery induce an adaptive response to cellular stress known as the integrated stress response (ISR; *Abbink et al., 2019*; *Ishimura et al., 2016*; *Spaulding et al., 2021*; *Terrey et al., 2020*). Canonical activation of the ISR is initiated by stress-activated protein kinases which phosphorylate eIF2α to globally attenuate bulk mRNA translation initiation, increase translation of uORF-containing mRNAs including transcription factors such as ATF4 and induce a stress response transcription program (*Costa-Mattioli and Walter, 2020*). We measured the expression of 10 ATF4-regulated ISR target genes to evaluate whether the ISR had been activated in Polr3a-tamKI cerebra. A two to sixfold upregulation was measured for seven of these genes, two others were induced at lower levels and one was unaffected (*Figure 3D*). These results point to altered translation in Polr3a-tamKI cerebra.

Bulk RNAseq was used to broadly profile the transcriptional changes in Polr3a-tamKI cerebra at P75. Differentially expressed (DE) genes (p adj. <0.05, log2FoldChange >|0.58|) were biased towards upregulation; 83% of DE genes were increased in the *Polr3a* mutant (*Figure 3E*, *Supplementary file 1*). We compared our DE genes with a list of 774 ISR genes that show ATF4-dependent induction following tunicamycin treatment of mouse embryo fibroblasts (*Torrence et al., 2021*) and found 192 genes upregulated in Polr3a-tamKI cerebra (*Figure 3E*, *Figure 3—figure supplement 1A*, *Supplementary file 1*). Among these were key ISR regulators, such as uORF-containing transcription factors (*Atf3*, *Atf5* and *Ddit3*, aka *Chop*), pro-apoptotic targets of *Ddit3* (*Chac1* and *Bbc3*), and negative ISR regulators (*Ppp1r15a*, aka *Gadd34*, and *Trib3*) that function to antagonize the response (*Figure 3E*).

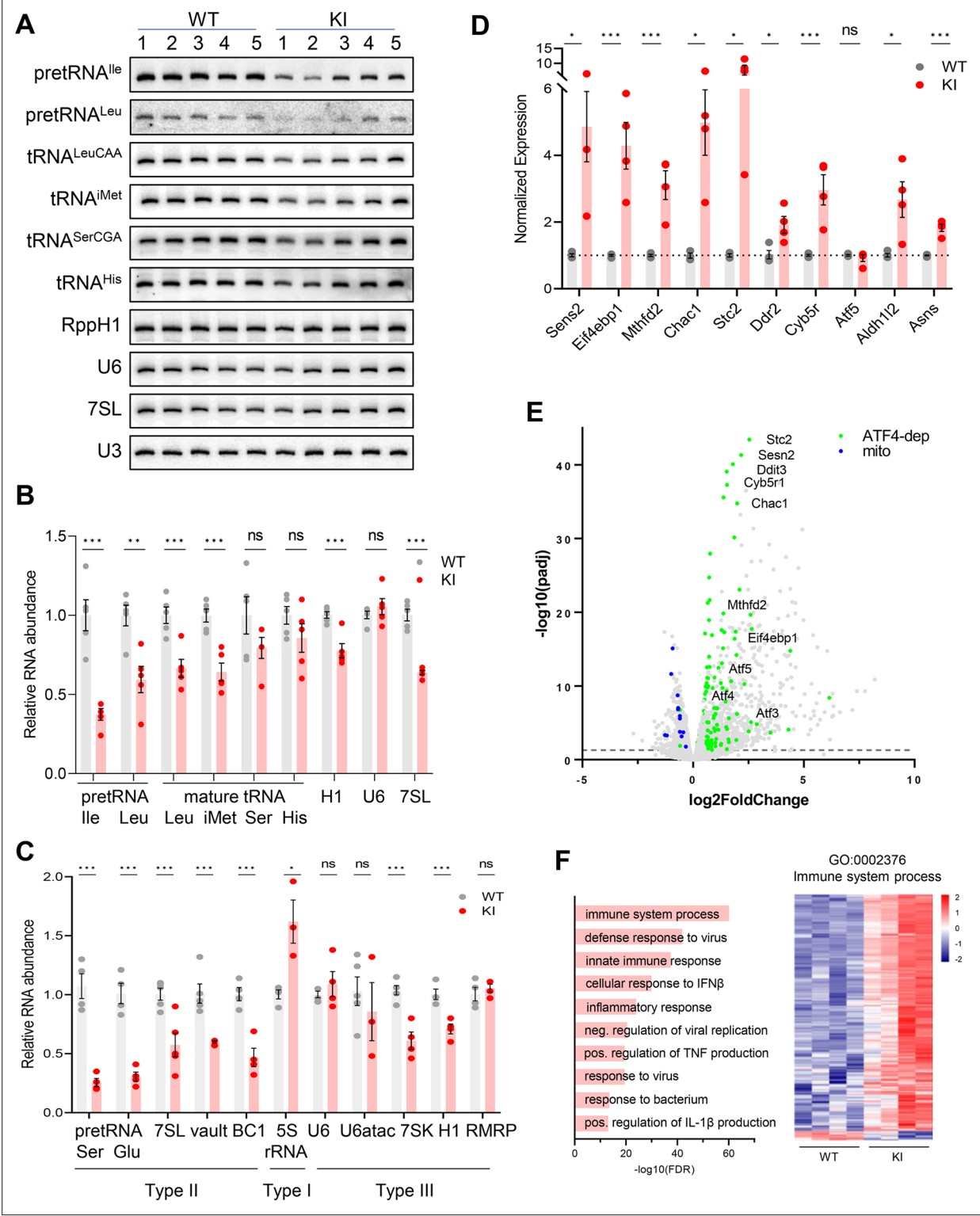

**Figure 3.** Transcriptome analysis of Polr3a-tamKI cerebra at P75. (**A**) Northern blot of Pol III transcripts. Precursor tRNAs (Ile-TAT, Leu-CAA), mature tRNAs (Leu-CAA, iMet, Ser-CGA, His), RPPH1, U6 and 7SL RNAs and the loading control U3 snRNA were detected by hybridization using transcript-specific oligonucleotide probes. All blots represent sequential probing of a single gel. The cropped images frame the relevant regions. (**B**) Quantitation of Pol III transcripts in panel A (mean ± SEM, n=5 biological replicates). (**C**) RT-qPCR of Pol III transcript abundance (mean ± SEM, n=3–6 biological replicates). (**D**) RT-qPCR of selected ATF4-regulated ISR genes (mean ± SEM, n=3–5 biological replicates). (**E**) Global gene expression quantified by RNA-seq. The volcano plot shows gene expression changes (KI/WT, n=4 biological replicates). Differentially expressed (DE) ATF4-regulated genes

*Figure 3 continued on next page*

*Figure 3 continued*

(green) (p-adj <0.05, log2FC >|0.58|) and mitochondrially encoded transcripts (blue). ATF4-regulated genes assayed in panel D and Ddit3/Chop are labeled. (**F**) Top ten GO bioprocesses among up-regulated DE genes are shown with a heatmap of Z-scores for DE genes annotated to the GO:0002376 immune system process. For all graphs: WT, gray bars and circles; KI, red bars and circles. ns, not significant; * p≤0.05; ** p≤0.01; *** p≤0.005.

The online version of this article includes the following source data and figure supplement(s) for figure 3:

**Source data 1.** PDF file containing original northern blots for *Figure 3A*, indicating the relevant bands and conditions.

**Source data 2.** Original files for northern analysis displayed in *Figure 3A*.

**Figure supplement 1.** Gene expression in WT and Polr3a-tamKI (KI) cerebra at P75.

**Figure supplement 2.** Pol III and Pol II transcript levels in various tissues from WT and Polr3a-tamKI (KI) mice at P75.

**Figure supplement 2—source data 1.** PDF file containing original northern blots for *Figure 3—figure supplement 2A*.

**Figure supplement 2—source data 2.** Original files for northern analysis displayed in *Figure 3—figure supplement 2A*.

Interestingly, expression of the transcription factor ATF6 and its canonical ERAD targets (*Pdia4*, *Pdia6*, *Edem1*, *Grp78* aka *Hspa5*, *Grp94* aka *Hsp90b1*, *Xbp1*, and *Canx*) were not affected suggesting that ISR induction in the *Polr3a* mutant does not involve an ER stress-mediated Unfolded Protein Response (UPR, *Supplementary file 1*; *Hillary and FitzGerald, 2018*). This was confirmed by showing that the relative levels of inactive unspliced and active spliced forms of *Xbp1*, a critical transcription factor for the UPR, were unchanged (*Figure 3—figure supplement 1B*).

Metascape pathway enrichment of upregulated DE genes identified activation of the immune system and related processes as major functional categories (*Figure 3F*, *Supplementary file 1*). These data point to activation of microglia (see below) and the upregulation of signaling pathways that promote inflammation and cell death by multiple mechanisms (*Butovsky and Weiner, 2018*). For example, pyroptosis is indicated by the upregulation of caspases 1 and 4, the inflammasome sensor NLRP3 and its adapter protein, PYCARD, interleukin-1β and the plasma membrane pore-forming protein, Gasdermin D, among others (*Supplementary file 1*). The most enriched functional categories among genes down-regulated in Polr3a-tamKI cerebra were cholesterol biosynthesis and electron transport (*Supplementary file 1*). All mitochondrially encoded transcripts were also down-regulated in the *Polr3a* mutant (*Figure 3E*). In contrast, nuclear-encoded mitochondrial genes were mostly unaffected although 24 were found in the upregulated gene set (*Supplementary file 1*, *Figure 3—figure supplement 1C*). These observations suggested a loss of mitochondrial DNA. Indeed, qPCR analysis showed that the ratio of two mitochondrial genes (*Mt-Rnr2* and *Mt-Nd1*) to a nuclear gene (*Hk2*) was substantially reduced in Polr3a-tamKI cerebra (*Figure 3—figure supplement 1D*). Thus, the loss of mitochondrial genomes underlies the reduction in mitochondrially encoded RNA detected by RNAseq. *Hmgcr* and *Sqle* were among numerous sterol biosynthetic genes down-regulated in Polr3a-tamKI cerebra (*Figure 3—figure supplement 1E*, *Supplementary file 1*). These enzymes function in rate-limiting steps in de novo cholesterol synthesis, suggesting dysfunctional cholesterol production. Cholesterol synthesis in the brain occurs predominantly in astrocytes and oligodendrocytes, accounts for almost one quarter of total body cholesterol and cannot be compensated by the periphery since cholesterol uptake is blocked by the blood-brain barrier (*Saher and Stumpf, 2015*). Astrocytes synthesize cholesterol for export to neurons while oligodendrocytes (OLs) synthesize 70–80% of brain cholesterol that resides in myelin (*Barber and Raben, 2019*; *Li et al., 2022*). Impairment of cholesterol biosynthesis in either cell type can result in cognitive and behavioral phenotypes (*Li et al., 2022*) and thus may contribute to the behavioral defects noted in Polr3a-tamKI mice.

## Changes in cell-type-specific gene expression reflect the loss of neurons and oligodendrocytes and activation of microglia

We queried the Polr3a-tamKI DE genes against consensus sets of mouse brain cell type gene expression signatures for OLs, neurons, astrocytes, microglia and endothelial cells (*McKenzie et al., 2018*). Twenty-one percent of Polr3a-tamKI DE genes (p-adj <0.05) could be assigned to one of these five cell types with microglia and neurons constituting the bulk of the cell-type-specific genes (*Supplementary file 2*). OL and neuronal gene expression was predominantly down-regulated suggesting a reduction in these cell populations while microglia were predominantly elevated (*Figure 4A–C*, *Supplementary file 2*). Immunohistochemistry confirmed that these changes in cell type gene expression signatures correlated with changes in cell-specific protein markers (*Figure 4D–H*). Myelination in

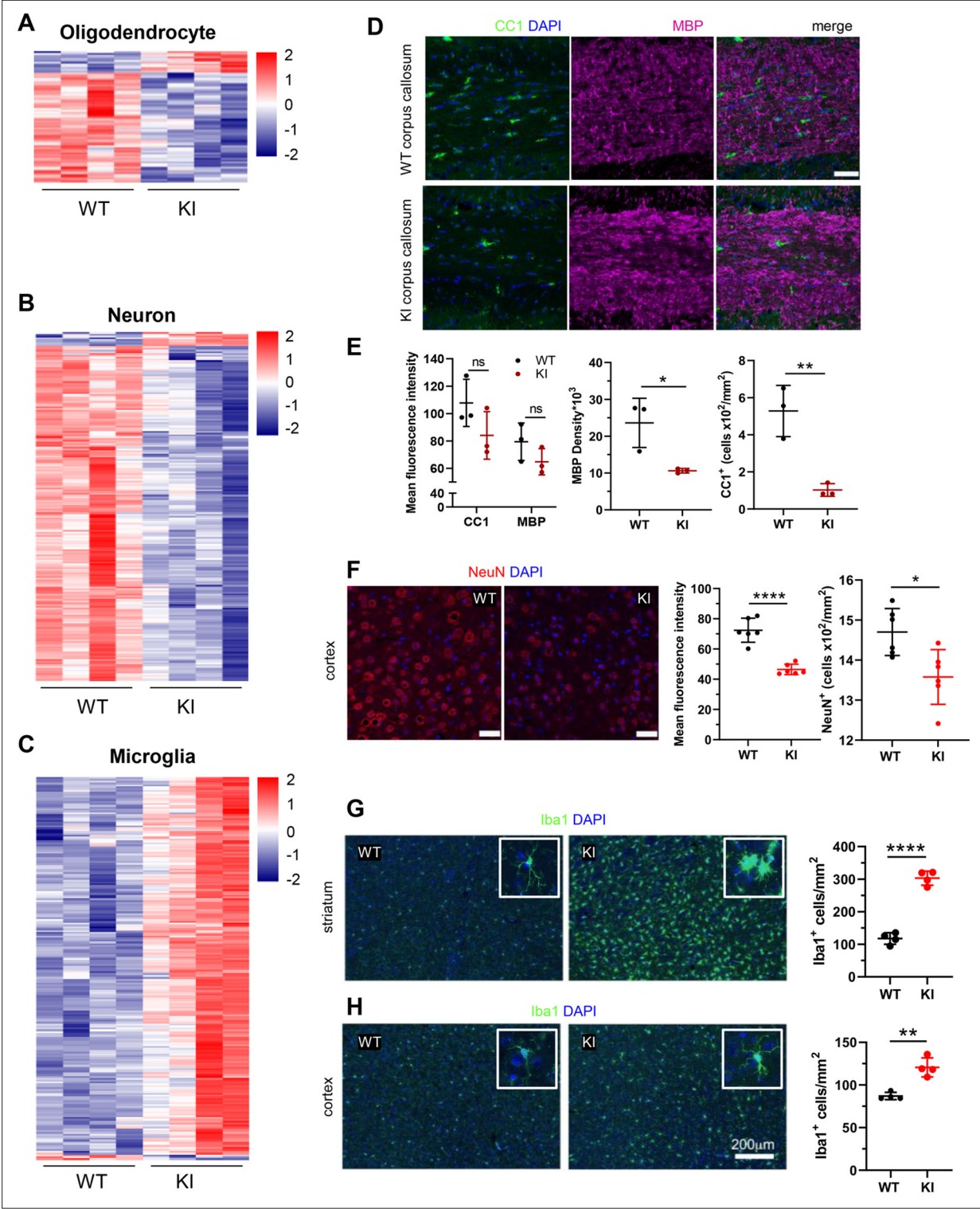

**Figure 4.** Cell-type-specific changes in Polr3a-tamKI cerebra at P75. (**A–C**) Oligodendrocyte (**A**), neuron (**B**), and microglia (**C**) cell type-specific DE genes. Heatmaps show Z-scores of DE genes identified in the top 500 mouse cell-type-specific genes defined in *McKenzie et al., 2018*. (**D–E**) Immunostaining of oligodendrocytes at the midline of the corpus callosum. Scale bar, 40 µm. Mean fluorescence intensities of MBP and CC1, MBP staining density and CC1 cell counts represent mean ± SD, n=3. (**F**) NeuN immunostaining of neurons in the motor cortex. Scale bar, 40 µm. Mean fluorescence intensities and cell counts represent mean ± SD of bilaterally symmetric regions (n=3). (**G–H**) Iba1 staining of microglia in the striatum (**G**) and cerebral cortex (**H**). Insets show larger magnifications of individual microglia. Note the striking shortening of processes and enlargement of the cell body in the striatum. Scale bar, 200 µm. Cell counts represent mean ± SD of bilaterally symmetric regions, (n=2). For all graphs: WT, black; KI, red; ns, not significant; * p≤0.05; ** p≤0.01; *** p≤0.005; **** p<0.0001.

the mouse begins at birth and is largely complete by weaning, although it continues with decreasing frequency through adulthood (*Hammelrath et al., 2016*; *Nishiyama et al., 2021*). Thus, under our treatment regimen, substantial myelination has occurred by P28 when the *Polr3a* mutation is introduced. This, together with the long half-life of myelin (*Barnes-Vélez et al., 2023*) contributes to a high background level of myelin. Accordingly, the mean fluorescence intensity of myelin basic protein (MBP) staining in the corpus callosum of adult Polr3a-tamKI mice was not significantly diminished relative to WT. However, the density of MBP staining was reduced as was the number of OLs staining positive with the CC1 antibody (*Figure 4D and E*). In the cerebral cortex, the mean fluorescence intensity of NeuN staining and the number of NeuN positive neuronal nuclei was lower in Polr3a-tamKI mice (*Figure 4F*) consistent with the decrease in neuron-specific gene expression. Additionally, we observed reactive microglia in Polr3a-tamKI cerebra, as evidenced by the increased cell density and altered morphology (enlarged cell bodies and shorter, thicker processes) of Iba1-stained cells in the cortex and particularly in the striatum (*Figure 4G and H*). Together, these data indicate that behavioral deficits, induction of the ISR, loss of mitochondrial genomes, an elevated immune response and neurodegeneration are causally linked to the loss of Pol III transcription in Polr3a-tamKI cerebra.

To assess whether the changes observed in Pol III and Pol II transcripts in P75 cerebra occurred in other tissues, we examined cerebellum, heart, kidney, and liver from the same animals. The levels of three precursor tRNAs in the cerebellum, heart, kidney were reduced ~threefold, comparable to the changes seen in cerebra (*Figure 3—figure supplement 2A*). However, in the liver, where the efficiency of CAGGCre-ER recombination is lower (*Figure 1—figure supplement 1C*; *Hayashi and McMahon, 2002*), only one of the three precursor tRNAs was reduced (*Figure 3—figure supplement 2A*). The levels of two mature tRNAs were variably affected, being unchanged in liver and cerebella, trending lower in heart and reduced in kidney to ~70% of WT (*Figure 3—figure supplement 2A*). Among other Pol III transcripts, tissues other than cerebra showed few changes: *7SL* and *Rmrp* RNA levels were reduced in cerebella but otherwise no reductions were detected (*Figure 3—figure supplement 2B*). Similarly, there was little indication of an ISR response in the tested tissues other than cerebra (*Figure 3—figure supplement 2C*). These data imply that the diminished activity of Pol III in these tissues is above the threshold necessary for ISR induction.

## A reduction in Pol III transcription and mature tRNA levels precedes induction of innate immune and integrated stress responses

To separate causal changes in Pol III activity from secondary stress responses that could down-regulate Pol III transcription, we profiled both the Pol II and Pol III transcriptomes of WT and Polr3a-tamKI cerebra at P42, two weeks after induction of recombination. Similar to the P75 time point, Polr3a-tamKI pre-tRNA levels were reduced ~threefold and three out of four mature tRNAs were reduced to ~60% of the WT level (compare *Figure 5A–C* with *Figure 3A–C*). No Pol III transcripts other than tRNAs were decreased at P42 (*Figure 5C*). Thus, decreased production of precursor and mature tRNA is the earliest detectable Pol III defect in Polr3a-tamKI cerebra.

Analysis of the ATF4-regulated ISR gene panel revealed that fewer genes were induced in Polr3a-tamKI cerebra at P42 and the response was less robust than at P75 (compare *Figure 5—figure supplement 1A* and *Figure 3D*). *Sesn2*, *Mthfd2*, *Chac1*, *Ald1l2,* and *Asns* were up-regulated twofold or less indicating partial activation of the ISR. Bulk RNAseq showed fewer significant DE genes at P42 (181 genes) with 68% up-regulated and minimal overlap with the P75 DE gene set (*Figure 5D*, *Figure 5—figure supplement 1B*, *Supplementary file 3*). Although no GO terms were enriched, the P42 DE gene set contained a small number of ISR genes (20 genes), the majority of which are ATF4-regulated and showed increased expression (*Figure 5—figure supplement 1B*, *Supplementary file 3*). Similarly, a small number of up-regulated genes were associated with the immune response GO bioprocess (21 genes at P42 versus 582 genes at P75, *Supplementary file 3*). Additionally, the expression of mitochondrially-encoded genes was unaffected at P42, as was mitochondrial genome content (*Figure 5—figure supplement 1C*, *Supplementary file 3*). Consistent with these data at P42 and in contrast to the findings at P75, Iba1-staining did not reveal any increase in the density of microglia in the cerebral cortex or the striatum (*Figure 5—figure supplement 1F, G*). Together these data indicate a progression of stress and immune response programs with time in Polr3a-tamKI cerebra.

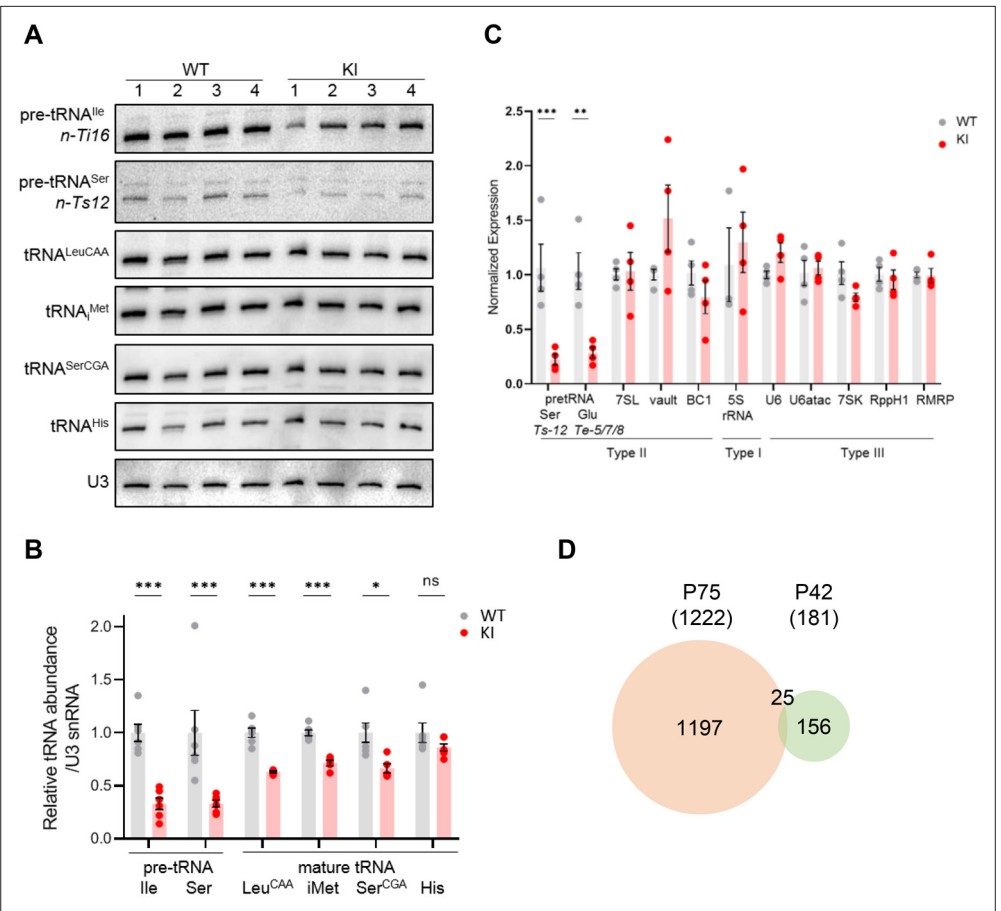

**Figure 5.** Transcriptome analysis of Polr3a-tamKI cerebra at P42. (**A**) Northern blots of Pol III transcripts. Precursor tRNAs (Ile-TAT, Ser-CGA), mature tRNAs (Leu-CAA, iMet, Ser-CGA, His) and U3 snRNA were detected as in *Figure 3A*. (**B**) Quantitation of Pol III transcripts in panel *A*. Mean ± SEM, n=5 biological replicates. (**C**) RT-qPCR of Pol III transcript abundance. Mean ± SEM, n=3–5 biological replicates. (**D**) Venn diagram of the overlap between P75 and P42 DE genes (p-adj <0.05, log2FC >|0.58|).

The online version of this article includes the following source data and figure supplement(s) for figure 5:

**Source data 1.** PDF file containing original northern blots for *Figure 5A*, indicating the relevant bands and conditions.

**Source data 2.** Original files for northern analysis displayed in *Figure 5A*.

**Figure supplement 1.** Gene expression and Iba1 staining in adolescent WT and Polr3a-tamKI (KI) cerebra.

## The *Polr3a* mutation reduces total cytoplasmic tRNA levels and alters the tRNA profile

The effect of the *Polr3a* mutation on the tRNAome in P42 cerebra was assessed by QuantM-tRNA-seq (*Pinkard et al., 2020*). This method exploits the conserved 3'-CCA end of mature tRNAs using a splint-ligation approach to prepare DNA libraries for sequencing. Prior to sample workup, a synthetic tRNA spike-in based on *E. coli* tRNA^Gln was added for normalization of tRNA reads to total input RNA. Since mitochondrially encoded mRNAs and mitochondrial DNA content were comparable between WT and Polr3a-tamKI cerebra at P42 (*Figure 5—figure supplement 1C*, *Supplementary file 3*), mitochondrially encoded tRNAs were also used for normalization. Total nuclear-encoded tRNA reads normalized with either or both standards were similar and significantly lower in Polr3a-tamKI cerebra suggesting an ~25% reduction in total tRNA abundance (*Figure 6A*, *Figure 5—figure supplement 1D, E*). Differential expression (DE) analysis indicated that the majority of tRNA isodecoders (i.e. tRNAs that share the same anticodon but have different body sequences) were significantly lower in the *Polr3a* mutant (58%, 108/187 tRNAs with p-adj <0.05, *Supplementary file 5*). DE was observed across the entire read

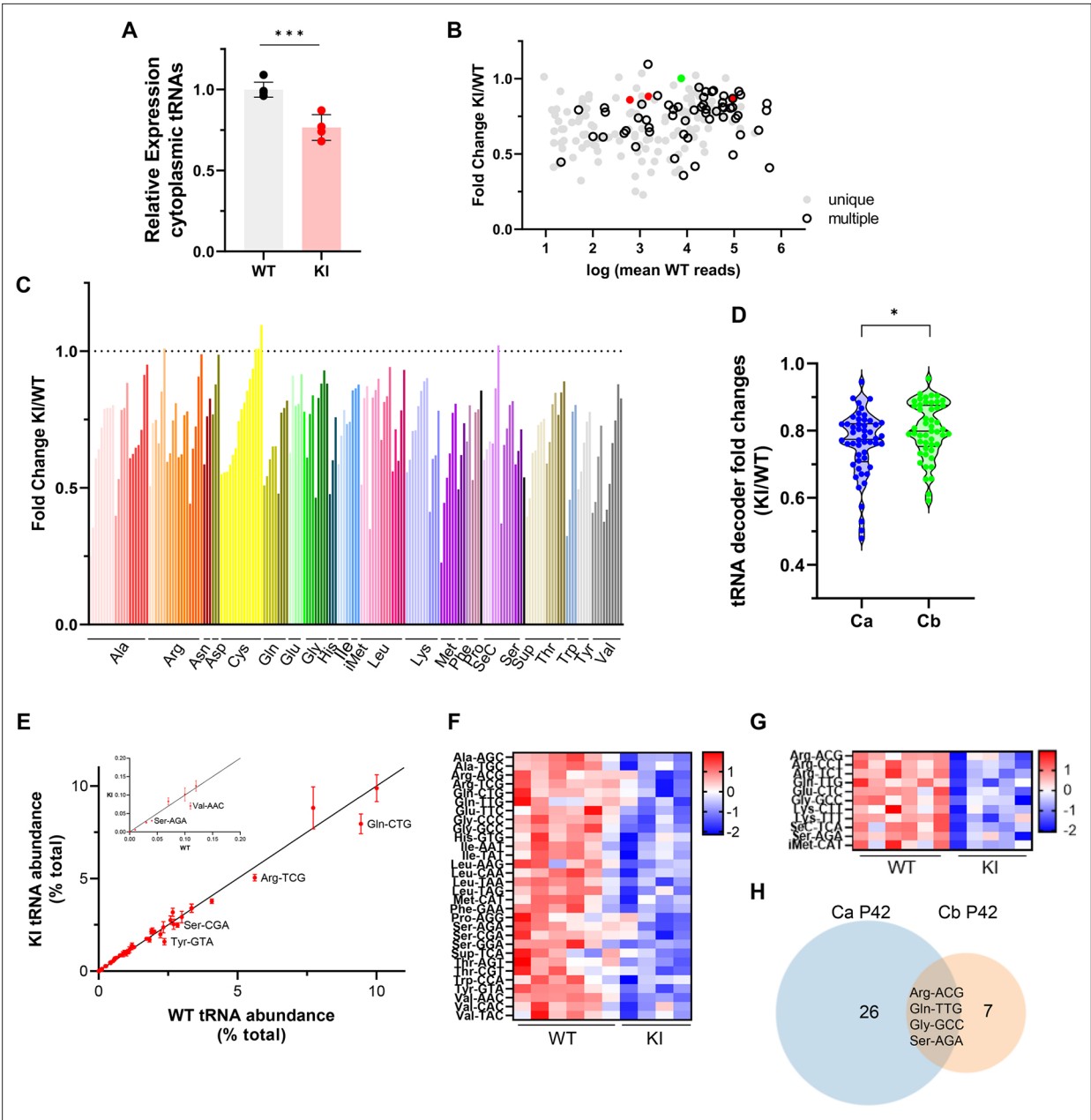

**Figure 6.** Cytosolic tRNA abundance in Polr3a-tamKI cerebra at P42. (**A**) Total cytoplasmic tRNA reads in WT and KI cerebra, normalized to the sum of spike-in and mitochondrially-encoded tRNA reads are expressed relative to the mean WT value. Mean ± SD, KI n=4, WT n=6 biological replicates, p=0.0003. (**B**) tRNA fold change (KI/WT) is plotted against Log mean WT reads. Symbols show tRNAs encoded by unique loci (gray), identical tRNAs encoded by multiple loci (hollow black), iMet tRNAs (red) and *n-TRtct5* (green). (**C**) tRNA fold change (KI/WT) for all decoder families. Individual tRNAs are ordered from most to least fold change and grouped by codon recognition (tRNA decoder family). The amino acid for each tRNA decoder family is indicated. (**D**) Violin plots of tRNA decoder fold changes (KI/WT) for cerebra (Ca) and cerebella (Cb) at P42. tRNA reads were summed for each tRNA decoder family and normalized to spike-in and mitochondrial tRNA reads, p=0.0250. For Ca, n=4 (KI) and n=6 (WT) biological replicates and for Cb, n=5 (KI) and n=6 (WT) biological replicates. (**E**) The cytoplasmic tRNA profile for KI cerebra is plotted against the WT profile. tRNA decoder reads are expressed as a percentage of their respective total cytoplasmic decoder pool, mean ± SEM. tRNA decoders that are significantly lower in the KI compared to WT (p≤0.05) fall below the regression line and are labeled. The inset shows tRNA decoders representing <0.2% of the total tRNA pool. (**F**) DE tRNA decoders in cerebra (Ca). (**G**) DE tRNA decoders in cerebella (Cb). Heatmaps represent Z scores of normalized read counts for significant DE genes (p-adj <0.05). (**H**) Venn diagram of the overlap between DE tRNA decoders in Ca and Cb (p-adj <0.05).

The online version of this article includes the following figure supplement(s) for figure 6:

**Figure supplement 1.** Gene expression in WT and Polr3a-tamKI (KI) cerebella at P42.

depth which spans 5 orders of magnitude and included tRNAs whose sequences could be uniquely mapped as well as redundantly encoded tRNAs (**Figure 6B**). Compared to most elongator tRNAs, the three tRNAiMet isodecoders were less dramatically affected, reduced to 86–88% of the WT level. The level of *n-TRtct5* (aka *tRNA-Arg-TCT-4–1*) an abundant neuron-enriched tRNA that is mutated in the C57BL/6 J background and causes synthetic phenotypes when combined with ribosome rescue factor mutants (**Ishimura et al., 2014**; **Terrey et al., 2020**), was unchanged in the *Polr3a* mutant. Thus, the cerebral phenotypes of Polr3a-tamKI mice are unlikely to result from synthetic effects of the *n-TRtct5* and *Polr3a* mutations.

Grouping tRNA isodecoders into their isoacceptor families illustrates that most members of each family were reduced in the *Polr3a* mutant (**Figure 6C**). Indeed, when tRNA read counts were combined into their respective decoder (i.e. anticodon) groups, all decoders were reduced in the *Polr3a* mutant, and for 30 decoders (60%), the reductions were significant (**Figure 6F**, **Supplementary file 5**). To assess how these changes affect the composition of the tRNA population (i.e. the tRNA profile), each decoder was plotted as a percentage of the total tRNA population for the *Polr3a* mutant versus WT (**Figure 6E**). This revealed changes in the relative proportions of some tRNAs. For example, tRNAVal(AAC) and tRNATyr(GTA) were proportionally less abundant than tRNAHis(GTG), and seven other tRNAs, respectively. These changes coupled with the overall decrease in tRNA abundance have the potential to change the kinetics of tRNA binding to the ribosome (decoding speed) as well as the kinetic competition with near-cognate tRNAs for ribosome binding. The most severely affected decoders were tRNAVal(AAC) (to 48% of the WT level), tRNATyr(GTA) (to 50%), tRNASup(TCA) (to 53%), tRNASer(AGA) (to 63%) and tRNAGln(CTG) (to 64%). In mouse, tRNASer(AGA) and tRNAVal(AAC) decode 41% and 42% of their respective codons (accounting for 3rd base wobble), tRNAGln(CTG) decodes 75% of glutamine codons and tRNATyr(GTA) decodes 100% of tyrosine codons. These findings suggest that the decrease in tRNA decoders is likely to negatively affect the tRNA decoding potential at the earliest stage of disease pathogenesis.

## All tRNA isodecoder families are reduced in Polr3a-tamKI cerebellum but few decoders are affected

An initial analysis of the Pol III transcriptome in Polr3a-tamKI cerebella at P42 revealed comparable findings to the cerebra with decreases in several precursor and mature tRNAs and no effects on other Pol III transcripts (compare **Figure 6—figure supplement 1A, B** with **Figure 5B and C**). QuantM-tRNA-seq indicated that total tRNA abundance was reduced in cerebella by ~23% in the *Polr3a* mutant (**Figure 6—figure supplement 1D**), similar to cerebra, and the expression of tRNA isodecoders was broadly decreased (**Figure 6—figure supplement 1E, F**). However, a comparative DE analysis showed that fewer tRNAs (30 vs 108) and fewer tRNA decoders (11 vs 30) achieved statistical significance in Polr3a-tamKI cerebella compared to cerebra (**Figure 6G and H**, **Supplementary file 6**) and the distribution of fold change values spanned a narrower range and was shifted towards smaller differences relative to WT (**Figure 6D**). In addition, the tRNA profile was minimally altered in cerebella (**Figure 6—figure supplement 1G**). These differential effects on the tRNA population in cerebella versus cerebra are consistent with bulk RNA-seq data from cerebella which showed few significant DE genes (13 genes, p-adj <0.05, log2Fold Change >|0.58|, **Supplementary file 4**) and limited overlap with cerebral DE genes (**Figure 6—figure supplement 1C**). The minimal effect of the *Polr3a* mutation on the Pol II transcriptome of cerebella at P42 and the absence of ISR induction at P75 suggests that any reduction in tRNA decoding potential is insufficient to cause significant pathology in this tissue.

## Discussion

Patients with *Polr3*-related disease express *Polr3* mutations in all cells during development and throughout postnatal life yet exhibit predominantly CNS phenotypes. To gain insight into this neural sensitivity, we introduced a pathogenic *Polr3a* mutation into mice using a ubiquitously expressed tamoxifen-inducible Cre recombinase and examined the effects of reduced Pol III transcription in various tissues in a largely post-developmental state. The paucity of gross phenotypes identified by histopathology across many *Polr3a* mutant tissues reflects that cell-specific requirements for a minimum threshold of Pol III transcription have mostly been met. At the molecular level, our analysis of heart, kidney, cerebella, and cerebra showed that Pol III transcription, as reported by pre-tRNA

levels, was substantially reduced at P75, 6 weeks after induction of the mutation. These changes were paralleled by smaller decreases in mature tRNAs (*Figure 3—figure supplement 2A*) suggesting that tRNA half-lives are either much longer than expected in WT mice (*Nwagwu and Nana, 1980*) and/or that tRNA turnover is decreased in the *Polr3a* mutant. Surprisingly, the levels of non-tRNA Pol III transcripts were not decreased in heart or kidney and only *7SL* and *Rmrp* RNAs were lower in cerebella at P75. In contrast, reductions among non-tRNA transcripts were widespread in cerebra at the same age although two RNAs remained unaffected (*Figure 3—figure supplement 2B*). Given the levels of recombination in these tissues (which were higher in cerebella, kidney, and heart than in cerebra, *Figure 3—figure supplement 2C*), the results point to a particularly high sensitivity of cerebra to the *Polr3a* mutation and a remarkable insensitivity of many non-tRNA transcripts in cerebella, kidney and heart to the reduction in Pol III activity. This insensitivity of non-tRNA transcripts may result from tissue-specific differences in the recruitment of the mutant Pol III to its genes, differences in the stability or nuclease accessibility of the RNA in specific ribonucleoprotein complexes, the ability of some Pol III promoters to be transcribed by RNA polymerase II (*Dergai et al., 2018*; *Gao et al., 2018*; *James Faresse et al., 2012*) or other effects that remain to be defined.

The higher sensitivity of cerebra versus cerebella, heart, and kidney to the *Polr3a* mutation was apparent at both P42 and P75. In cerebella, only 13 DE genes were identified at P42 whereas in cerebra the number of DE genes was more than an order of magnitude higher and included a nascent ATF4-dependent ISR that was absent in cerebella at this age. Moreover, while the induction of ISR genes in cerebra increased at P75, there was no appreciable ISR induction in cerebella, heart, and kidney. Together, these data suggest different thresholds of sensitivity to defects in Pol III transcription in different tissues and/or cell populations. Indeed, within the cerebra, the more robust activation of microglia in the striatum versus the cortex suggests significant differences in sensitivity to the *Polr3a* mutation between these regions. Thus, different thresholds of Pol III activity may be needed to support the functions of these regions, for example the continuing maturation of cortical thickness and myelination that occurs from P28 into adulthood (*Hammelrath et al., 2016*; *Nishiyama et al., 2021*) versus the functions other cells such as medium spiny neurons that comprise ~95% of the neurons in the striatum (*Tepper and Bolam, 2004*).

In contrast to the cerebra, the apparent insensitivity of the cerebella to the *Polr3a* mutation may reflect that neurogenesis and synapse formation is largely complete by P21, before introduction of the mutation (*Zeiss, 2021*). However, it remains possible that more subtle phenotypes such as dendritic pruning, which continues in the cerebellum through 2 months of age, may be affected (*Leto et al., 2016*). Of note, our developmental oligodendrocyte lineage model (Polr3a-Olig2KI mice), which showed reduced growth and neurological deficits along with hypomyelination in the cerebra and spinal cord (*Merheb et al., 2021*), also did not show cerebellar phenotypes. Together, these observations suggest that oligodendrocyte lineage cells in the cerebellum have a different threshold for minimal Pol III activity through development and maturation than in other regions of the brain.

The sensitivity of the mouse exocrine pancreas to the *Polr3a* mutation was unexpected given the absence of reports on acinar dysfunction associated with *Polr3*-related disease but is consistent with the knowledge that a *Polr3b* mutation disrupts development of the exocrine pancreas and intestine in zebrafish (*Yee et al., 2007*). Thus, our finding may inform undiagnosed digestive issues in the patient population. The unique codon usage of mouse and human pancreatic acinar cells, which differs from most other cell types and is skewed by the codon composition of highly expressed secreted proteins may underlie this phenotype (*Gao et al., 2022*). Alternatively, the *Polr3a* pancreatic phenotype may reflect unique differences in the development of rodent versus human pancreata: The increase in the size of mouse pancreas from birth to adulthood results largely from an increase (~19-fold) in the volume of acinar cells (*Anzi et al., 2018*). This hypertrophy of murine acinar cells is attributed to their higher biosynthetic rate and tetraploid genome content. In contrast, postnatal growth of human pancreas involves an increase in cell number, with acinar cells maintaining a constant volume (*Anzi et al., 2018*).

Pol III dysfunction and the reduction in the cerebral tRNA population at P42 coincides with behavioral deficits and precedes substantial downstream alterations in the Pol II transcriptome, which include induction of an innate immune response (IR) and an ISR, and indicators of neurodegeneration (i.e. activation of cell death pathways and loss of mitochondrial DNA). These findings suggest a causal role for the lower tRNA abundance and/or altered tRNA profile in disease progression. Whether

the decrease in the abundance of other Pol III transcripts in the cerebrum at later times contributes to disease progression is an open question. These changes may represent secondary effects of IR/ISR signaling or cell death stressors and may be occurring in cells that have already committed to a pathway of programmed cell death.

Recent studies have shown that decreased levels of a single cytoplasmic tRNA isoacceptor or deletion/mutation of a single nuclear-encoded tRNA gene is sufficient to alter translation and cause neuronal dysfunction: Dominant mutations in the glycyl-tRNA synthetase gene associated with Charcot-Marie-Tooth disease CMT2D, cause sequestration of tRNA$^{Gly}$, neuropathy, impair neuronal translation and induce the ISR (*Spaulding et al., 2021*; *Zuko et al., 2021*). Mutation or deletion of the *n-TRtct5* isodecoder expressed in neurons, reduced the total tRNA$^{Arg}$(UCU) pool, caused ribosome pausing at AGA codons, ISR induction, defects in neuronal function and sensitization to loss of ribosome quality control proteins (*Ishimura et al., 2014*; *Kapur et al., 2020*; *Terrey et al., 2020*). Global deletion of the *n-TFgaa7* (aka *tRNA-Phe-1–1*) isoacceptor is sufficient to cause transcription and proteomic changes, neurological deficits, increased ribosome stalling, and altered protein expression (*Hughes et al., 2023*). In the cerebra of Polr3a-tamKI mice, the induction of an ATF4-dependent ISR at P42 and its enhancement at P75 is a clear indicator that translation kinetics have been impacted by the global decrease in tRNA abundance, which extends across most decoders and changes the tRNA profile. Accordingly, the *Polr3a* mutation is predicted to compromise translation efficiency and change the frequency and sites of ribosome pausing and/or stalling. These events can trigger ISR induction via protein misfolding or unresolved ribosome collisions (*Costa-Mattioli and Walter, 2020*; *Park et al., 2021*). We propose that in the context of *Polr3*-related disease, chronic ISR activation ultimately fails to restore cellular homeostasis, and thus leads to neurodegeneration through amplification of the IR and activation of cell death pathways (*Costa-Mattioli and Walter, 2020*).

## Materials and methods

**Key resources table**

| Reagent type (species) or resource | Designation | Source or reference | Identifiers | Additional information |
|---|---|---|---|---|
| Genetic reagent (*Mus musculus*) | C57BL/6J-*Polr3a*$^{tm1lwil}$ | *Merheb et al., 2021* | PMCID:PMC8501794 | |
| Genetic reagent (*M. musculus*) | B6.Cg-*Tg(CAG-cre/Esr1*5Amc/J)* | Jackson Laboratory | #004682, RRID:IMSR_JAX:004682 | |
| Genetic reagent (*M. musculus*) | B6.129(Cg)-*Gt(ROSA)$^{26Sortm4(ACTB-tdTomato,-EGFP)Luo}$/J* | Jackson Laboratory | #007676, RRID:IMSR_JAX:007676 | |
| Antibody | Anti-MBP, rat monoclonal | Abcam | #ab7349, RRID:AB_305869 | 1:100 |
| Antibody | Anti-APC (CC1), mouse monoclonal | Millipore | #OP80, RRID:AB_2057371 | 1:20 |
| Antibody | Anti-NeuN, mouse monoclonal | Abcam | #ab104224, RRID:AB_10711040 | 1:100 |
| Antibody | Anti-Iba1, rabbit polyclonal | FUJIFILM Wako | #019–19741, RRID:AB_839504 | 1:250 |
| Antibody | Anti-mouse IgG, Alexa Fluor 488 goat polyclonal | Invitrogen | #A28175, RRID:AB_2536161 | 1:1000 |
| Antibody | Anti-mouse IgG2b Alexa-Fluor 568 goat polyclonal | Invitrogen | #A21144, *RRID*: AB_2535780 | 1:1000 |
| Antibody | Anti-rat IgG Alexa Fluor 633 goat polyclonal | Invitrogen | #A21094, *RRID*: AB_2535749 | 1:1000 |
| Antibody | Anti-rabbit IgG Alexa Fluor 488 goat polyclonal | Invitrogen | #A11008 *RRID*: AB_143165 | 1:1000 |
| Commercial assay or kit | Adult brain dissociation kit, mouse | Miltenyi | #130-107-677 | |
| Commercial assay or kit | Lightcycler 480 SYBR Green I Master mix | Roche LifeScience | #04707516001 | |

*Continued on next page*

*Continued*

| Reagent type (species) or resource | Designation | Source or reference | Identifiers | Additional information |
|---|---|---|---|---|
| Chemical compound, drug | ProLong diamond antifade with DAPI | Invitrogen | #36962 | |
| Chemical compound, drug | Paraformaldehyde 32% | Electron Microscopy Sciences | #15,714 S | |
| Chemical compound, drug | Tamoxifen | Millipore Sigma | #T5648 | |
| Chemical compound, drug | Corn oil | Millipore Sigma | #C8267 | |
| Chemical compound, drug | SuperScript III | Thermo Fisher | #18080051 | |
| Chemical compound, drug | SuperScript IV | Thermo Fisher | #18091050 | |
| Chemical compound, drug | TRIzol Reagent | Thermo Fisher | #15596018 | |
| Chemical compound, drug | RNaseOUT | Invitrogen | #10777019 | |
| Chemical compound, drug | T4 polynucleotide kinase | New England Biolabs | #M0201 | |
| Software, algorithm | CaseViewer v2.4 software | 3DHistech | RRID:SCR_017654 | |
| Software, algorithm | Viewer software | Biobserve | RRID:SCR_014337 | |
| Software, algorithm | Volocity v5.3 | Perkin Elmer | RRID:SCR_002668 | |
| Software, algorithm | Prism v9.0 | GraphPad Software | RRID:SCR_002798 | |
| Software, algorithm | ImageQuant v5.2 | GE Healthcare | RRID:SCR_014246 | |
| Software, algorithm | ImageJ v1.53 | Github | RRID:SCR_003070 | |
| Software, algorithm | R studio v1.3.10.93 | Posit | RRID:SCR_000432 | |
| Software, algorithm | DEseq2 | Bioconductor | RRID:SCR_015687 | |

## Mouse husbandry

All experiments involving mice were performed using protocols (00001373 and 00001376) approved by the Institutional Animal Care and Use Committee of the Albert Einstein College of Medicine (AECOM). Mice were housed in a barrier facility at 22 °C on a 12 hr light/dark cycle with constant access to food (PicoLab Mouse Diet 20) and water. C57BL/6J-*Polr3a*$^{tm1lwil}$ mice were described previously (*Merheb et al., 2021*) and were bred to B6.Cg-*Tg(CAG-cre/Esr1*5Amc/J)* mice (CAGGCre-ER, #004682) and to B6.129(Cg)-*Gt(ROSA)26Sor*$^{tm4(ACTB-tdTomato,-EGFP)Luo}$/J mice (tdTomato-EGFP, #007676). Whole-body inducible *Polr3a* mutant mice and Cre-minus WT controls were obtained by breeding male mice heterozygous for both the floxed *Polr3a* allele and Cre recombinase to female mice homozygous for the floxed *Polr3a* allele. The breeding of mice to assess recombination used females homozygous for the tdTomato-EGFP reporter. Experimental animals (knockin mutants and WT controls) were treated with tamoxifen (6 mg/40 g body weight, five injections i.p.) every other day starting at P28. Unless otherwise indicated, all experiments were performed with male and female mice and the data were pooled for analysis.

## Recombination efficiency

Single-cell suspensions of dissected cerebrum and cerebellum were prepared using an adult brain dissociation kit (Miltenyi). Flow analysis was performed on a LSRII-U flow cytometer (BD Biosciences) with single color controls to establish gating prior to sample analysis (*Merheb et al., 2021*). For liver, heart and kidney, fixed, sectioned tissue was mounted with ProLong diamond antifade with DAPI (Invitrogen) and imaged using a 3D Histech p250 high-capacity slide scanner. Manual cell counts were performed in ImageJ (five sections/tissue, >150 cells/section).

## Purkinje cell counts

Images of Nissl-stained sagittal sections of cerebellum were viewed using CaseViewer software for manual counting of Purkinje cells in lobes III, IV/V, VI/VII, and VIII. The open polygon tool was used to measure the length of the Purkinje layer for calculating Purkinje cell density.

## Histology

Necropsy and staining of tissues with H&E, LFB, and Nissl was performed in the Histopathology and Comparative Pathology Facility at AECOM. Images were acquired using a 3D Histech p250 high-capacity slide scanner.

## Mouse behavior

Behavioral studies were conducted during the light cycle using a Behavioral Spectrometer (BiObserve) equipped with video-tracking, infrared beams and a vibration-sensitive floor to record motion in the X, Y, and Z planes and to measure the frequency and magnitude of vibrations in a 40 cm$^2$ arena. Loco-motor activity was tracked and stereotyped behaviors were measured algorithmically using Viewer Software (BiObserve) (*Brodkin et al., 2014*). Specifically, locomotor classes of behavior include Run, Trot, Walk, and Still (not moving) metrics. Locomotor activity - open field-like behavior is defined by Track Length (total distance travelled in the arena). Exploratory classes of behavior include Rearing and Orientation behaviors. Rearing movements include Rear Climb (e.g. trying to climb walls), Rear Bob (moving body in an up/down vertical motion) and Rear Sniff (head movement in a nodding arc) activities. Orientation movements include Orient Shuffle (repositioning of all 4 feet and facing a different direction), Orient Creep (repositioning 1–3 steps more or less in the same direction) and Orient Sniff (moving less than a body length with no head movement) motions. Grooming behaviors are defined by Scratches (involving the hind legs) and grooming of specific body parts (Genital, Tummy, Back, Leg, Head, Face, Nose, and Paw). Anxiety-like behavior is defined by Center visits and Center duration which score the number and duration of visits to a 15 cm$^2$ central area of the arena and Center track (total distance travelled in the central area). Individual mice were placed in the instrument for a total of 9 min and recorded in 3-min intervals (3 recordings in total). Behaviors are reported after 6 min (two recordings) and 9 min (all three recordings). Approximately, equal numbers of mice of both genders (five WT females, four WT males and three KI females, two KI males) were examined at weekly intervals on postnatal days 42, 49, 56, and 63. The data for each genotype at each age were combined for analysis. Significance was calculated by multiple t tests and one-way ANOVA in GraphPad Prism.

## Immunohistochemistry

Mice at P63-P66 were transcardially perfused with 4% paraformaldehyde (PFA), brains were removed and postfixed overnight in 4% PFA at 4 °C before storage in 25% sucrose. Brains were embedded in OCT medium, cut into 10-μm-thick matched sections and stored at –20 °C. Sections were post-fixed in 4% PFA for 10 min at room temperature and rinsed in PBS 3x5 min before permeabilization and blocking in 1% Triton X-100, 10% goat serum, 10% BSA, PBS for 2 hr at room temperature prior to staining. Slides were incubated overnight at 4 °C with primary antibodies, MBP (Abcam Cat# ab7349, RRID:AB_305869; 1:100), CC1 (Millipore Cat# OP80, RRID:AB_2057371; 1:20) or NeuN (Abcam Cat# ab104224, RRID:AB_10711040; 1:100) in 0.1% Triton X-100, 1% goat serum, 10% BSA, PBS and then washed with PBS 3x5 min. Secondary antibodies (1:1000) were conjugated to Alexa-Fluor 488 (Invitrogen A28175), Alexa-Fluor 568 (Invitrogen A21144) or Alexa Fluor 633 (Invitrogen A21094). For Iba1 staining, antigen retrieval (0.025% Tween-20 in 10 mM citrate buffer pH 6.0) and non-specific blocking (10% donkey serum) was performed before incubation overnight with primary antibody (Wako Chemicals RRID:AB_839504; 1:250) and detection with IgG conjugated to Alexa 488 (Invitrogen A11008). After washing in PBS, slides were mounted with ProLong diamond antifade with DAPI (Invitrogen). Images were captured using a 3D Histech P250 High Capacity Slide Scanner and Z-stack images were acquired using a Leica SP8 inverted DMi8 confocal microscope (40 X N.A.1.3).

## Immunofluoresence and quantitation

LIF micrographs were analyzed using Volocity software. Composite images were generated and then split into individual channels. The threshold for the green, red and magenta channels was set uniformly across all sections according to signal intensity. Mean fluorescent intensity and area values were reported directly from Volocity software. CC1 stained cells were counted at the midline of the corpus callosum and NeuN stained cells were counted in the motor cortex. Colocalization of CC1 and MBP signals was interrogated in the merged image throughout the Z-stack and confirmed in the individual split channels. Quantification of Iba1 straining was performed manually in areas covering ~1.3 mm$^2$ of the cortex and striatum.

## Nucleic acid preparation for RNA and DNA analysis

Tissues (50–100 mg, freeze-clamped and flash frozen in liquid nitrogen) were homogenized into TRIzol Reagent. RNA was reprecipitated before quantification. DNA recovered using the TRIzol DNA isolation protocol was used in mitochondrial genome content assays (*Quiros et al., 2017*). Total RNA (2 µg) was used for cDNA synthesis with SuperScript III (SSIII, Invitrogen). qPCR followed the Lightcycler 480 SYBR Green I Master mix product manual for 384 multiwell plates (Roche). All primers are listed in *Supplementary file 7*. Each sample was run in triplicate with four to six biological replicates. ΔΔCt values were calculated using two reference genes (GAPDH, β-actin and/or γ-tubulin) (*Taylor et al., 2019*).

## Northern blotting

Total RNA (5 µg) was resolved by denaturing polyacrylamide electrophoresis before electrophoretic transfer to Nytran N or SPC membranes (Cytiva), and hybridization to [$^{32}$P]-end-labeled oligonucleotide probes at 37 °C (*Bonhoure et al., 2015*). Transcripts detected by phosphorimaging were quantified using ImageQuant software, normalized to U3 snRNA, and expressed as a fraction of the mean WT value. Oligonucleotide probes are listed in *Supplementary file 7*.

## tRNA-sequencing and differential expression analysis

Total RNA was mixed with a synthetic spike-in tRNA based on *E. coli* tRNAGln (*Supplementary file 7*). The spike-in tRNA was denatured and refolded (80 °C 2 min with a 0.1 °C/s ramp to 20 °C in 0.5 mM EDTA), and then added to 2.5 µg of total RNA from P42 cerebra (0.375 ng spike-in) or P42 cerebella (0.750 ng spike-in) prior to tRNA library preparation. The RNA mixture was deacylated (37 °C for 45 min in 20 mM Tris-HCl pH 9.0, RNaseOUT [1 µg/µL, Invitrogen]) and then 3'-end dephosphorylated and 5'-end phosphorylated using T4 polynucleotide kinase (NEB). tRNA library preparation used SuperScript IV following the QuantM-tRNA-seq protocol (*Pinkard et al., 2020*). Library multiplexing, sequencing (Illumina NextSeq500, single-end 150 bp reads) and mapping was performed by the Center for Epigenomics/Computational Genomics Core at AECOM. Library reads were trimmed and mapped, as specified previously (*Pinkard et al., 2020*), to a custom look-up table containing the high confidence set of nuclear-encoded cytosolic tRNAs (gtRNAdb grcm38/mm10), mitochondrially encoded tRNAs (RNAcentral) and the spike-in tRNA sequence. tRNA reads > 10 were used for further analyses. Decoder families were created by summing reads for all tRNAs with the same anticodon sequence. tRNAseq raw read counts for tRNAs and tRNA decoder families were normalized to the mitochondrial tRNA and spike-in tRNA read counts using the estimateSizeFactors control gene function in DESeq2 (*Love et al., 2014*). Differential expression and significance were determined using the likelihood ratio test (LRT) in DESeq2. Data visualization and plotting was performed in Prism (v9, GraphPad).

## RNAseq

Poly(A) selected mRNA libraries were generated from total RNA (RINs >8.9, Agilent Bioanalyzer), sequenced (Illumina PE150) and processed by Novogene. Differential expression and significance was determined using DESeq2 in R studio for all read counts >10 using default settings and the Wald test. DE was defined as adjusted p-value <0.05. Downstream data visualization and plotting were performed using pheatmap in R and Prism (v9, GraphPad).

## Statistical analysis

Statistical analyses were performed using Prism (v9 GraphPad) or Excel. Two-tailed unpaired Student's t tests were used for experiments with two conditions. Behavioral testing used multiple unpaired t tests and one-way ANOVAs. In molecular analyses, all samples represent biological replicates (no samples were pooled). Significance was determined at $p < 0.05$. * p<0.05, ** p<0.01, *** p<0.001. Data are presented as mean ± SD or ± SEM as indicated.

# Acknowledgements

We thank Hillary Guzik for her help with confocal image acquisition and Volocity analysis, Dr. Robert A Dubin for computational support in mapping tRNAseq reads and Jinghang Zhang, MD

for support with flow cytometry. This work was supported by a National Institutes of Health grant R21 NS123730 (to IMW and RDM) and RO1 NS129951 (to IMW). Additional support was provided by the Rose F Kennedy Intellectual and Developmental Disabilities Research Center (IDDRC), which is funded through a center grant from the Eunice Kennedy Shriver National Institute of Child Health & Human Development (NICHD U54-HD090260), by an Albert Einstein Cancer Center core support grant (P30-CA013330) and Analytical Imaging facility support grants (1S10OD026852 and 1S10OD023591).

## Additional information

### Funding

| Funder | Grant reference number | Author |
|---|---|---|
| National Institutes of Health | R21 NS123730 | Robyn D Moir<br>Ian M Willis |
| National Institutes of Health | RO1 NS129951 | Ian M Willis |

The funders had no role in study design, data collection and interpretation, or the decision to submit the work for publication.

### Author contributions

Robyn D Moir, Conceptualization, Data curation, Formal analysis, Funding acquisition, Validation, Investigation, Visualization, Writing - original draft, Writing – review and editing; Emilio Merheb, Conceptualization, Formal analysis, Validation, Investigation, Visualization, Writing - original draft, Writing – review and editing; Violeta Chitu, Formal analysis, Validation, Investigation, Visualization, Writing – review and editing; E Richard Stanley, Supervision, Writing – review and editing; Ian M Willis, Conceptualization, Formal analysis, Supervision, Funding acquisition, Validation, Investigation, Visualization, Writing - original draft, Writing – review and editing

### Author ORCIDs

E Richard Stanley ![ORCID] https://orcid.org/0000-0002-2910-2065
Ian M Willis ![ORCID] https://orcid.org/0000-0001-6599-2395

### Ethics

All experiments involving mice were performed using protocols (00001373 and 00001376) approved by the Institutional Animal Care and Use Committee of the Albert Einstein College of Medicine (AECOM).

Joint public review: https://doi.org/10.7554/eLife.95314.3.sa1
Author response https://doi.org/10.7554/eLife.95314.3.sa2

## Additional files

### Supplementary files

- MDAR checklist
- Supplementary file 1. RNAseq analysis of WT and Polr3a-tamKI cerebra at P75.
- Supplementary file 2. Mouse cell type-specific DE genes in Polr3a-tamKI cerebra at P75.
- Supplementary file 3. RNAseq analysis of WT and Polr3a-tamKI cerebra at P42.
- Supplementary file 4. RNAseq of WT and Polr3a-tamKI cerebella at P42.
- Supplementary file 5. tRNAseq P42_Ca. tRNAseq results and analysis of P42 cerebra.
- Supplementary file 6. tRNAseq analysis of WT and Polr3a-tamKI cerebella at P42.
- Supplementary file 7. Oligonucleotide sequences.

## Data availability

All data generated or analyzed during this study are included in the manuscript and supporting files. Sequencing data have been deposited in GEO under accession number GSE246162.

The following dataset was generated:

| Author(s) | Year | Dataset title | Dataset URL | Database and Identifier |
|---|---|---|---|---|
| Moir RD, Merheb E, Chitu V, Stanley ER, Willis IM | 2023 | Molecular basis of neurodegeneration in a mouse model of Polr3-related disease | https://www.ncbi.nlm.nih.gov/geo/query/acc.cgi?acc=GSE246162 | NCBI Gene Expression Omnibus, GSE246162 |

The following previously published dataset was used:

| Author(s) | Year | Dataset title | Dataset URL | Database and Identifier |
|---|---|---|---|---|
| Torrence ME, MacArthur MR, Hosios AM, Valvezan AJ, Asara JM, Mitchell JR, Manning BD | 2020 | The mTORC1-mediated activation of ATF4 promotes protein and glutathione synthesis | https://www.ncbi.nlm.nih.gov/geo/query/acc.cgi?acc=GSE158605 | NCBI Gene Expression Omnibus, GSE158605 |

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
