## [Editor Report · eLife Assessment]

This study provides **important** insights into the mechanistic basis of neurological manifestations of RNA polymerase III-related disease by creating a mutant mouse to dissect transcriptional changes. The data provide **compelling** evidence for disease progression initiated by a global reduction in tRNA levels leading to integrated stress and innate immune responses and neuronal loss. The work will be of interest to those engaged in the study of chromosome biology, developmental biology and neurodegeneration.

---

## [Referee Report · Joint public review]

Summary:

The authors present an intriguing investigation into the pathogenesis of Pol III variants associated with neurodegeneration. They established an inducible mouse model to overcome developmental lethality, administering 5 doses of tamoxifen to initiate the knock-in of the mutant allele. Subsequent behavioral assessments and histological analyses revealed potential neurological deficits. Robust analyses of the tRNA transcriptome, conducted via northern blotting and RNA sequencing, suggested a selective deleterious effect of the variant on the cerebrum, in contrast to the cerebellum and non-cerebral tissues. Through this work, the authors identified molecular changes caused by Pol III mutations, particularly in the tRNA transcriptome, and demonstrated its relative progression and selectivity in brain tissue. Overall, this study provides valuable insights into the neurological manifestations of certain genetic disorders and sheds light on transcripts/products that are constitutively expressed in various tissues.

Strengths:

The authors utilize an innovative mouse model to constitutively knock in the gene, enhancing the study's robustness. Behavioral data collection using a spectrometer reduces experimenter bias and effectively complements the neurological disorder manifestations. Transcriptome analyses are extensive and informative, covering various tissue types and identifying stress response elements and mitochondrial transcriptome patterns. Additionally, metabolic studies involving pancreatic activity and glucose consumption were conducted to eliminate potential glucose dysfunction, strengthening the histological analyses.

Comments on revised version from expert Editor #1:

The authors in the revised manuscript have effectively responded to all of the comments and suggestions raised by both reviewers. Overall, I find the revised version to be an important contribution to the field and the strength of evidence supporting the work's claims to be compelling.

Comments on revised version from expert Editor #2:

The authors have responded constructively to all the comments in the first round of reviews and clarified many issues in the manuscript. The current report represents a significant advance.

Comments on revised version from Reviewer #2:

The authors should include their clarifications of all concern raised by reviewer #2 (mentioned in the previous weaknesses) in the main text. They should consider including point #2 to point #10 in the main text (discussion section). The should highlight limitations of this study in discussion.

Also, they should clearly state that deciphering brain area specific behavioural deficits is beyond the scope of the manuscript with appropriate justification mentioned in the rebuttal letter.

I still do not agree with the author to state that "brain region-specific sensitivities to a defect in Pol III transcription". The changes are global and also not restricted to brain. Authors may consider restating this sentence. It is obvious that transcription defects related to tRNA production will lead to alteration in whole body physiology.

---

## [Author Response]

The following is the authors’ response to the original reviews.

**Reviewer #1 (Public Review):**
Summary:Moir, Merheb et al. present an intriguing investigation into the pathogenesis of Pol III variants associated with neurodegeneration. They established an inducible mouse model to overcome developmental lethality, administering 5 doses of tamoxifen to initiate the knock-in of the mutant allele. Subsequent behavioral assessments and histological analyses revealed potential neurological deficits. Robust analyses of the tRNA transcriptome, conducted via northern blotting and RNA sequencing, suggested a selective deleterious effect of the variant on the cerebrum, in contrast to the cerebellum and non-cerebral tissues. Through this work, the authors identified molecular changes caused by Pol III mutations, particularly in the tRNA transcriptome, and demonstrated its relative progression and selectivity in brain tissue. Overall, this study provides valuable insights into the neurological manifestations of certain genetic disorders and sheds light on transcripts/products that are constitutively expressed in various tissues.Strengths:The authors utilize an innovative mouse model to constitutively knock in the gene, enhancing the study's robustness. Behavioral data collection using a spectrometer reduces experimenter bias and effectively complements the neurological disorder manifestations. Transcriptome analyses are extensive and informative, covering various tissue types and identifying stress response elements and mitochondrial transcriptome patterns. Additionally, metabolic studies involving pancreatic activity and glucose consumption were conducted to eliminate potential glucose dysfunction, strengthening the histological analyses.Weaknesses:The study could have explored identifying the extent of changes in the tRNA transcriptome among different cell types in the cerebrum. Although the authors attempted to show the temporal progression of tRNA transcriptome changes between P42 and P75 mice, the causal link was not established. A subsequent rescue experiment in the future could address this gap.Nonetheless, the claims and conclusions are supported by the presented data.

We thank Reviewer 1 for their thoughtful review and commentary. We appreciate the reviewer’s finding that our “claims and conclusions are supported by the presented data.”

We note that our findings on the temporal progression of transcriptional changes between P42 and P75 apply to both the Pol II and Pol III transcriptomes. Importantly, in the case of Pol III, only precursor and mature tRNAs are affected at P42 whereas at P75, numerous other Pol III transcripts are also changed. We therefore attribute the changes in tRNA as being causal in disease initiation since this is the earliest direct consequence of the *Polr3a* mutation.

To expand on the evidence demonstrating the progressive nature of *Polr3*-related disease in our mouse model, the revised manuscript includes new immunofluorescence data showing no change in microglial cell density in the cerebral cortex or the striatum at an early stage in the disease (Supplementary Fig. S6F, G). This is in striking contrast to the findings at later times (P75) where the number of microglia increased significantly in the *Polr3a* mutant and exhibit an activated morphology (Fig. 4G,H).

We agree with the reviewer that it will be interesting in the future to assess the impact of the *Polr3a* mutation in different neural cell types and to explore opportunities for suppressing disease phenotypes.

**Reviewer #2 (Public Review):**
Summary:The study "Molecular basis of neurodegeneration in a mouse model of Polr3 related disease" by Moir et.al. showed that how RNA Pol III mutation affects production, maturation and transport of tRNAs. Furthermore, their study suggested that RNA pol III mutation leads to behavioural deficits that are commonly observed in neurodegeneration. Although, this study used a mouse model to establish theses aspects, the study seems to lack a clear direction and mechanism as to how the altered level of tRNA affects locomotor behaviour. They should have used conditional mouse to delete the gene in specific brain area to test their hypothesis. Otherwise, this study shows a more generalized developmental effect rather than specific function of altered tRNA level. This is very evident from their bulk RNA sequencing study. This study provides some discrete information rather than a coherent story. My enthusiasm for publication of this article in eLife is dampened considering following reasons mentioned in the weakness.

Reviewer 2’s summary contains two misstatements:

Moir et.al. showed that how RNA Pol III mutation affects production, maturation and transport of tRNAs.

Our experiments document the effect of a neurodegenerative disease-causing mutation in RNA polymerase III on the Pol III transcriptome with a particular focus on the tRNAome (i.e. the mature tRNA population). Experiments on the maturation and transport of tRNA were not performed as there was no indication that these processes might be negatively impacted at the earliest time point (P42). Additional comments about tRNA maturation and export are provided under points 8 and 9 (see below).

The study seems to lack a clear direction and mechanism as to how the altered level of tRNA affects locomotor behaviour.

This comment misstates the purpose of our study while overlooking the important results. As stated in the abstract, our goal was to develop “a postnatal whole-body mouse model expressing pathogenic *Polr3a* mutations to examine the molecular mechanisms by which reduced Pol III transcription results primarily in central nervous system phenotypes.”

Accordingly, our work provides the first molecular analysis of RNA polymerase III transcription in an animal model of *Polr3*-related disease. The novelty and importance of the findings, as stated in the abstract, include the discovery that a global reduction in tRNA levels (and not other Pol III transcripts) at an early stage in the disease precedes the frank induction of integrated stress and innate immune responses, activation of microglia and neuronal loss at later times. These later events readily account for the observed neurobehavioral deficits that collectively include risk assessment, locomotor, exploratory and grooming behaviors.

Strengths:The study created a mouse model to investigate role of RNA PolIII transcription. Furthermore, the study provided RNA seq analysis of the mutant mice and highlighted expression specific transcripts affected by the RNA PolIII mutation.Weaknesses:(1) The abstract is not clearly written. It is hard to interpret what is the objective of the study and why they are important to investigate. For example: "The molecular basis of disease pathogenesis is unknown." Which disease? 4H leukodystrophy? All neurodegenerative disease?

We have modified the abstract to more clearly frame the objective of the study and its importance as reflected in the title “Molecular basis of neurodegeneration in a mouse model of *Polr3*-related disease”. We hope the reviewer will agree that the fourth sentence of the abstract, unchanged from the initial submission, clearly outlines the objective of the study.

(2) How cerebral pathology and exocrine pancreatic atrophy are related? How altered tRNA level connects these two axes?

It is not known how cerebral pathology and exocrine pancreatic atrophy are related beyond their shared Pol III dysfunction in our mouse model of *Polr3*-related disease. We anticipate that altered tRNA levels connect these two axes. Indeed, the pancreas and the brain are both known to be highly sensitive to perturbations affecting translation (Costa-Mattioli and Walter, 2020 Science doi: 10.1126/science.aat5314). Changes to the tRNA population in the cerebrum and cerebellum of *Polr3a* mutant mice were extensively documented in the manuscript (e.g. Figs. 3, 5 and 6). We also found reduced tRNA levels in the pancreas of the mutant mice but did not report these findings due to the absence of a stable reference transcript in total RNA from the atrophied pancreatic tissue, even at the earliest time point examined (P42).

(3) Authors mentioned that previously observed reduction mature tRNA level also recapitulated in their study. Why this study is novel then?

Our study reports the novel finding that a pathogenic *Polr3a* mutation causes a global reduction in the steady state levels of mature tRNAs, i.e. the levels of all tRNA decoders were reduced with the vast majority these reaching statistical significance (Fig. 6D and 6F). In the introduction we refer to several studies that examined the effect of pathogenic *Polr3* mutations on the levels of Pol III-derived transcripts. We noted that these studies examined only a small number of Pol III transcripts in CRISPR-Cas9 engineered cell lines, patient-derived fibroblasts and patient blood. Thus, no study until now has tested for or reported a global defect in the abundance of mature tRNAs in any model of *Polr3*-related disease. Moreover, no previous study of Polr3-related disease has analyzed Pol III transcript levels in the brain or in any other tissue.

(4) It is very intuitive that deficit in Pol III transcription would severely affect protein synthesis in all brain areas as well as other organs. Hence, growth defect observed in Polr3a mutant mice is not very specific rather a general phenomenon.

While we agree with the simple assumption that a “deficit in Pol III transcription likely would affect protein synthesis in all brain areas as well as other organs”, this turned out not to be the case. In fact, a novel finding of our study is that not all *Polr3a* mutant tissues show a translation stress response despite reduced Pol III transcription and reduced mature tRNA levels. This implies that in some tissues the reduction in tRNA levels caused by the *Polr3a* mutation is not sufficient to affect protein synthesis, at least to a point where the Integrated Stress Response is induced. The underlying basis for the growth deficit has not been defined in this work. However, we noted in the discussion that a growth defect was previously seen in mice where expression of the *Polr3a* mutation was restricted to the Olig2 lineage. In the present postnatal whole-body inducible model, we anticipate that the diminished growth of the mice results from a combination of hormonal and nutritional deficits caused by cerebral and pancreatic dysfunction.

(5) Authors observed specific myelination defect in cortex and hippocampus but not in cerebellum. This is an interesting observation. It is important to find the link between tRNA removal and myelin depletion in hippocampus or cortex? Why is myelination not affected in cerebellum?

We agree that the specific myelin defect observed in the cortex and hippocampus, but not the cerebellum, is an interesting observation. Pol III dysfunction in this model and reduced tRNA levels are common to both cerebra and cerebella, yet the pathological consequences differ between these regions. While we do not know why this is the case, the cells that oligodendrocytes support in these regions are functionally different. We suggest in the discussion that subtle defects in oligodendrocyte function in the cerebellum may be uncovered using more sensitive or specific assays than the ones we have employed to date. In addition, consistent with our findings in other tissues where Pol III transcription and tRNA levels are reduced but phenotypes are lacking, we suggest that oligodendrocytes in the cerebellum may have a different minimum threshold for Pol III activity than in other regions of the brain.

(6) How was the locomotor activity measured? The detailed description is missing. Also, locomotion is primarily cerebellum dependent. There is no change in term of growth rate and myelination in cerebellar neurons. I do not understand why locomotor activity was measured.

We used a behavioral spectrometer with video tracking and pattern-recognition software to quantify ~20 home cage-like behaviors, including locomotor activity, as part of our phenotypic characterization of the mice. This experimenter-unbiased approach reported several metrics of locomotion, specifically, total Track length (the total distance traveled in the instrument), Center Track length and the time spent running (Run Sum) and standing still (Still Sum) in a longitudinal study (Figs. 2A-C and Supplemental Fig. S3A-C). The Materials and Methods section on mouse behavior has been amended to provide a detailed description of these experiments.

locomotion is primarily cerebellum dependent

While we agree that the cerebellum plays a critical role in balance and locomotion, regions of the cerebrum that are affected in our mice, including the primary motor cortex and the basal ganglia (Fig. 4), also have important roles in locomotor activity and control.

(7) The correlation with behavioural changes and RNA seq data is missing. There a number of transcripts are affected and mostly very general factors for cellular metabolism. Most of them are RNA Pol II transcribed. How a Pol III mutation influences RNA Pol II driven transcription? I did not find differential expression of any specific transcripts associated with behavioural changes. What is the motivation for transcriptomics analysis? None of these transcripts are very specific for myelination. It is rather a general cellular metabolism effect that indirectly influences myelination.

The differentially expressed mRNAs identified in our RNAseq analysis at P75 reflect both direct and secondary consequences of dysfunctional Pol III transcription on Pol II transcription. These effects can be achieved by multiple mechanisms. Induction of the Integrated Stress Response (ISR) due to insufficient tRNA can be considered a direct consequence of diminished Pol III transcription on Pol II transcription. An example of a secondary response is the activation of microglia and the innate immune response (which is known to accompany prolonged activation of the ISR), and the loss of neurons and oligodendrocytes. These changes are documented in Figs. 3 and 4. Importantly, loss of neurons, activated microglia and reduced oligodendrocyte numbers are each readily reconciled with changes in behavior.

None of these transcripts are very specific for myelination

The RNAseq data at P75 indicates only a modest reduction in oligodendrocyte-specific gene expression (as defined by single-cell RNAseq studies of purified cell populations, Mackenzie et al., 2018 Sci. Rep. doi: 10.1038/s41598-018-27293-5). Despite this, some oligodendrocyte-specific transcripts with well-known roles in myelination were down-regulated in the *Polr3a* mutant (e.g. Plp1, Mog and Mobp). In addition, steroid synthesis pathway transcripts involved in the production of cholesterol, an abundant and essential component of myelin, were also downregulated (Supplementary Fig. S4E).

(8) What genes identified by transcriptomics analysis regulates maturation of tRNA? Authors should at least perform RNAi study to identify possible factor and analyze their importance in maturation of tRNA.

Of the many proteins involved in the maturation of tRNA (Phizicky and Hopper, 2023 RNA doi: 10.1261/rna.079620.123), RNAseq analysis at P75 identified only amino-acyl tRNA synthetases as being differentially-expressed (fold change >1.5, p adj. < 0.05, Table S1). These genes are canonical indicators of the ATF4-dependent Integrated Stress Response and their upregulation is widely interpreted as an attempt to restore efficient translation. In addition, our analysis of Pol III transcripts at P75 identified a reduction in the level of RppH1 (Fig. 3C), the RNA component of RNase P, which removes the 5’ leader of precursor tRNAs. However, at P42, there was no effect on RppH1 abundance, or the expression of amino-acyl tRNA synthetase genes (Fig. 5C and Table S3). Thus, an RNAi study to identify and analyze a possible factor involved in the maturation of tRNA is neither warranted nor relevant to the current body of work.

(9) What factors are influencing tRNA transport to cytoplasm? It may be possible that Polr3a mutation affect cytoplasmic transport of tRNA. Authors should study this aspect using an imaging experiment.

Our analysis of tRNA populations in this study employed total cellular RNA and thus reflect the abundance of mature tRNA from all cellular compartments. We have not assessed whether the reduction in tRNA abundance caused by the *Polr3a* mutation alters the dynamics of tRNA transport from the nucleus to the cytoplasm. However, we consider it highly unlikely that the *Polr3a* mutation would have a significant effect on cytoplasmic transport of tRNA. Imaging experiments along these lines are beyond the scope of the current study.

(10) Does alteration of cytoplasmic level of tRNA affects translation? Author should perform translation assay using bio-orthoganal amino acid (AHA) labelling.

It is not known whether the reduced tRNA levels affect translation globally in the *Polr3a* mutant, but we predict that this may not be the case. Since tissues (heart and kidney) and brain regions (cerebrum and cerebellum) that share a decrease in tRNA abundance do not share activation of the Integrated Stress Response (a reporter of aberrant translation), we anticipate that effects on translation may be limited to specific regions or cell populations and to specific mRNAs within these cells. The current study provides the foundation for further work to address these questions.

**Reviewer #1 (Recommendations For The Authors):**
Below are a few comments, mostly regarding typographical errors, presentation, and clarity, that we believe would enhance this manuscript:On the heatmaps generated, it would be ideal to place "WT" before "KI," with "WT" on the left. This will maintain consistency with the rest of the manuscript, where "WT" conditions precede "KI" conditions, as observed in the bar graphs and dot plots.

All heatmaps have been remade with WT on the left and KI on the right to maintain consistency throughout the manuscript.

Authors mentioned in several instances (Discussion Pg 19 Line 2, for instance) the analysis of changes in the "Pol II transcriptome." Is this a typographical error?

The reference to the Pol II transcriptome is not a typographical error (Discussion Pg 19 Line2). Here and elsewhere in the manuscript, we are distinguishing between changes to the Pol III transcriptome and the timing of subsequent changes to the Pol II transcriptome. The text has been edited to clarify this relationship in several places.

(1) Introduction, Page 4, last paragraph.

Analysis of the Pol III transcriptome reveals a common decrease in pre-tRNA and mature tRNA populations and few if any changes among other Pol III transcripts across multiple tissues. Analysis of the Pol II transcriptome reveals activation of the integrated stress response in cerebra but not in other surveyed tissues.

(2) Results, page 8, 2nd paragraph

To investigate the molecular changes to Pol III transcript levels caused by the *Polr3a* mutation and any secondary effects on the Pol II transcriptome, we initially focused on the cerebra of adult mice at P75.

(3) Discussion, Page 19, second paragraph

Pol III dysfunction and the reduction in the cerebral tRNA population at P42 coincides with behavioral deficits and precedes substantial downstream alterations in the Pol II transcriptome, which include induction of an innate immune response (IR) and an ISR, and indicators of neurodegeneration (i.e., activation of cell death pathways and loss of mitochondrial DNA). These findings suggest a causal role for the lower tRNA abundance and/or altered tRNA profile in disease progression.

In supplementary figure 1, authors validated the expression of their systems using flow cytometry and observed a high level of recombination frequency in different tissue types. Can the flow cytometry data distinguish between cell types within the cerebrum (neurons/microglia/astrocytes)?

The flow cytometry experiments reported in Supplementary Fig. S1 used a dual tdTomato-EGFP reporter to assess recombination. The cerebral and cerebellar samples were gated on fluorescence from endogenous expression of tdTomato (red), EGFP (green) and DAPI (blue) staining. In principle, flow cytometry could be used to distinguish between cell types within the cerebrum (neurons/microglia/astrocytes). However, this would require (i) an antibody to a cell surface marker on the cell type of interest and (ii) a fluorescent probe conjugated to the primary antibody or a fluorescent secondary antibody that is spectrally well resolved from the emission spectra of tdTomato, eGFP and DAPI.

Results section 1: Is there any particular reason why P28 was chosen as the commencement of tamoxifen injection?

P28 was chosen so that any effect of the *Polr3a* mutation on development and differentiation would be limited in the tissues we examined.

Fig 1C: The number of asterisks does not match between the graph and the figure legend.

Fig. 1C has been corrected to match the number of asterisks in the graph and figure legend.

Results section 3:This section seemed a little brief, especially when compared to the depth of the succeeding sections. Authors can state in greater detail which behaviors were quantified. In S3A-C, my understanding is that the animals were placed in an open-field test. This procedure can be briefly mentioned in the methods, as well as in the main manuscript text.In the legends of S3, a bracket is missing for "(D-F)" on line 5. Additionally, the alignment of legends for each bar graph could be consistent for all graphs except under the condition of spatial constraint.

Detailed methods pertaining to the measurement and calculation of home cage-like behaviors reported by the behavioral spectrometer have been added to the Methods section on Mouse Behavior.

In the Results, Figs. S3A-C show anxiety-like behaviors which measure the number and duration of visits and the distance traveled in a 15 cm2 central area of the arena. Figs. 2A-C show locomotor behaviors including Tracklength, Run sum and Still sum. The open field-like behavior is reported as total Tracklength in the behavioral spectrometer, i.e. the total distance travelled in the arena. This is now more clearly described in the main manuscript and the Methods section. “overall locomotor activity was decreased in Polr3a-tamKI mice as indicated by the reduced track length at P42, P49, P56 and P63 (Fig. 2A).”

The legend of S3, now has the missing bracket "(D-F)" on line 5.

The legends within each bar graph are now consistent and aligned as much as spatial constraints allow.

Results section 4:Similar to our earlier questions for S1, is it possible to distinguish samples derived from different cell types (neurons/glia)? In figure 4, this is mainly done post-hoc, based on the known gene expression. Maybe the authors could discuss this small limitation? In Fig S4C, the color contrast for the heatmap legend needs to be corrected.

It is not possible to accurately distinguish different neural cell sub-types, such as different types of neurons, or different types of oligodendrocytes in bulk RNAseq. Hence, we have reported only high confidence correlations based on known gene expression signatures (Fig. 4). We discuss only the data for which we can draw confident conclusions. The heatmap and legend in Fig. S4C has been amended.

Results section 5:In figure S5A, the alignment of asterisk significance markers could be adjusted.

Asterisks have been realigned in Fig. S5A

**Reviewer #2 (Recommendations For The Authors):**
Methods Section should include detailed procedure.

A detailed description of the methods pertaining to the measurement and calculation of behaviors using the behavioral spectrometer has been added to the Methods section.

Statistical tests should have detailed information

Statistical tests are detailed in the Methods section “Statistical Analysis”. Additional details pertaining to calculations of behavioral data have been added to the “Mouse behavior” section of the Methods.